# Evaluating the World Models Used by Pretrained Learners

## Abstract

A common approach for assessing whether generative models develop world models is by studying the behavior of fixed models. However, many of the benefits of having a world model arise when transferring a model to new tasks (e.g. few-shot learning). In this paper, we ask: what does it mean to test if a *learner* has a world model embodied in it? We consider a simple definition of a true world model: a mapping from inputs to states. We introduce a procedure that assesses a learner's world model by measuring its inductive bias when transferring to new tasks. This inductive bias can be measured in two distinct dimensions: does a learner extrapolate to new data by building functions of state, and to what degree do these functions capture the full state? We use this procedure to study the degree to which pretrained models extrapolate to new tasks based on state. We find that models that perform very well on next-token prediction can extrapolate to new tasks with very little inductive bias toward state. We conclude by assessing the possibility that these models learn bundles of heuristics that enable them to perform well on next-token prediction despite preserving little of state.

## 1 Introduction

A growing body of research investigates whether large language models (LLMs) and other foundation models form internal representations of the data they're trained on (Abdou et al., 2021; Li et al., 2023). Methods that uncover world models from sequential data would be valuable in many settings: they could be used to uncover scientific breakthroughs in domains such as protein generation, genetics, and chemistry (Chowdhury et al., 2022; Benegas et al., 2023; Jablonka et al., 2024; Boiko et al., 2023).

One of the biggest advantages of a model with an implicit world model is effective few-shot learning; with a correct world model, the same model can be transferred to different but related tasks with minimal modifications. However, much of the literature studying world models has focused on assessing the outputs of a fixed model (Toshniwal et al., 2022; Vafa et al., 2024). A true world model should manifest not just in making valid predictions, but in how a system learns and adapts to new situations using a generalizable representation of the domain. This is particularly important because many of the purported benefits of world models—like few-shot learning and transfer—specifically arise from how models learn new tasks. Here, rather than studying a fixed model, we ask: what does it mean to test if a *learner* has a world model embodied in it?

We consider a simple definition of a world model: a real-world representation of inputs in a low-dimensional state space. Meanwhile, a learner is any procedure that takes a dataset and returns a model that relates inputs to outputs. We then propose a procedure to test if a learner has a given world model: when a learner is applied to a new dataset, to what degree does it learn functions of this low-dimensional representation? For a learner to rely on a given world model, every dataset it is applied to should only be a function of this low-dimensional representation of reality. This definition is not just abstract; a world model in language could correspond to underlying concepts, so a learner that has the correct concepts should have an inductive bias to learn new functions of these concepts.

We introduce two related definitions that capture properties for whether a learner uses the world model. The first definition is about whether a learner respects state. This corresponds to whether a learner's predictions of points across new datasets obey state structure. For example, if the state space corresponds to a board in the game Othello, the learner respects state if all sequences that map to the same board have the same prediction within any given dataset. However, respecting state

doesn't convey the full story; for example, a model can make predictions using very coarse functions of state, such as how many pieces a particular Othello board has. To provide a fuller picture, we also provide a definition of what it means for a learner to fully reconstruct state. Learners that use coarse functions of state will not satisfy this second definition.

We present a computationally efficient method for estimating these quantities. Our method involves repeatedly applying a learner to small amounts of data on random outputs that obey state and studying how it extrapolates. We then build a model that predicts these extrapolations as a function of state. We quantify two properties to measure the learner's world model: first, the learner's inductive bias toward state is how well the extrapolations can be predicted from state. Next, the degree to which the learner recovers full state is given by how predictable original states are from a shared representation that is predictive of extrapolations.

We use this procedure to study the extent to which pretrained models use world models when fine-tuning. We consider several applications where the true world model is known. In the first application, we study a setting where orbital data obeys the world model of Newtonian mechanics. We pretrain a transformer on trajectories of planetary motion and ask: can the model transfer to other tasks that rely on Newtonian mechanics? We show that while the model appears to obey Newtonian mechanics for the task it's trained on, our metrics reveal poor inductive bias. We show that instead of recovering a compact world model, the learner is relying on piecemeal heuristics; while Newton's law of gravity can be recovered when the model is fine-tuned on narrow kinds of transfer data, the model implies nonsensical laws when fine-tuned on more general sequences.

We also perform analogous exercises in two other areas where the true world model is known: lattice problems (Liu et al., 2022; Vafa et al., 2024) and Othello games (Li et al., 2023; Nanda et al., 2023b; Hazineh et al., 2023). On lattices, we find that sequence models have strong inductive biases toward true state. On Othello, we find smaller inductive biases toward state. By way of calibration, we also consider oracle models that are directly pretrained on state, in order to calibrate the degree to which a model's extrapolative properties are limited by architecture. These oracle benchmarks show that while simple recurrent models like RNNs (Elman, 1990) and LSTMs (Hochreiter, 1997) have about as strong an inductive bias as their respective oracles, there is a large gap for transformer (Vaswani et al., 2017) and Mamba (Gu & Dao, 2023; Dao & Gu, 2024) models. We next demonstrate the implications of these metrics; our inductive biases have a strong correlation with transfer performance across tasks.

Our results show that models, despite performing well on next-token prediction, can have poor transfer properties and low inductive bias towards state. We conclude by assessing the possibility that these models — instead of learning compact representations of world models — learn bundles of heuristics (Karvonen et al., 2024) that enable them to perform well on next token prediction despite having poor transfer properties for new problems.

## 2 FRAMEWORK

In this section, we lay out our framework for defining whether an algorithm learns from data using an underlying world model. Let $x \in \mathcal{X}$ denote some input and $y \in \mathcal{Y}$ denote some output. In our framework, the underlying world model is summarized by some state space $\Phi$ and a mapping $\phi \colon \mathcal{X} \to \Phi$ that associates each input with some state $\phi(x) \in \Phi$. An example is any pair $(x, y)$ and a dataset $D = \{(x_1, y_1), \ldots, (x_n, y_n)\}$ is any finite collection of examples. A dataset $D$ is *consistent* with the underlying world model if for any pair $(x, y), (x', y') \in D$ with $\phi(x) = \phi(x')$ then $y = y'$. When evaluating a learner against a world model, we assume the learner is applied to datasets that are consistent with the world model. Let $D^{\Phi}$ denote the collection of all consistent datasets.

A learning algorithm, when given a dataset $D$, returns a prediction function $\widehat{m}(\cdot; D)$ that relates inputs $x$ to outputs $y$. We next state two definitions that capture properties related to whether a learning algorithm uses a world model $\phi$. Let $P(\cdot)$ be some chosen distribution over the inputs with $x \sim P(\cdot)$.

**Definition 2.1.** The learning algorithm *respects state* $\Phi$ if for all $D \in \mathcal{D}^{\Phi}$ there exists some function $f(\cdot; D) \colon \Phi \to \mathcal{Y}$ such that $\widehat{m}(x; D) = f(\phi(x); D)$ for all $x \in \mathcal{X}$ with $P(x) > 0$.

In other words, the learning algorithm respects state if its learned prediction function returns the same predictions on inputs mapped to the same state by the world model. While this captures an

intuitive property, it is nonetheless a weak requirement. As an extreme case, consider a learning algorithm that returns a constant prediction function when applied to any dataset; this trivial learning algorithm mechanically respects state. To distinguish such cases, we introduce a second property.

**Definition 2.2.** Consider a learning algorithm that respects state $\Phi$. The learning algorithm *fully reconstructs* state if there exists no non-injective $r\colon \Phi \to \Phi$ such that $\widehat{m}(x; D) = f(r(\phi(x)); D)$ for all $x \in \mathcal{X}$ with $P(x) > 0$ and $D \in \mathcal{D}^{\Phi}$.

The learning algorithm fully reconstructs state if its predictions cannot be expressed by coarsening the state space of the underlying world model. If not, the learning algorithm partially reconstructs state.

## 2.1 Measuring Inductive Bias towards State and Partial Reconstruction of State

Definitions 2.1-2.2 are binary properties of a learning algorithm. We next introduce evaluation metrics to measure how far a learning algorithm is from respecting state and fully reconstructing state. For both metrics, it is useful to introduce the best approximation of a prediction model based on state as

$$s^*(\phi(x); D) := \arg\min_{s\colon \Phi \to \mathcal{Y}} \mathbb{E}_{x \sim P(\cdot)}\left[\ell(\widehat{m}(x; D), s(\phi(x))\right]. \tag{1}$$

We can decompose the returned prediction function as

$$\widehat{m}(x; D) = s^*(\phi(x); D) + \epsilon(x; D) \tag{2}$$

for $\epsilon(x; D) = \widehat{m}(x; D) - s^*(\phi(x); D)$. The function $s^*(\phi(x); D)$ can be thought of as the function of state that is closest to the learned model's predictions, and it will be useful for assessing how close a learner is to respecting and fully reconstructing state.

First, Definition 2.1 implies that for any dataset $D \in \mathcal{D}^{\Phi}$, $\bar{\ell}(D) :=$ $\mathbb{E}_{x \sim P(\cdot)}[\ell(\widehat{m}(x; D), s^*(\phi(x); D))] = 0$. Consequently, for any chosen distribution $Q(\cdot)$ over datasets $D \in \mathcal{D}^{\Phi}$, it follows that if the learning algorithm respects state, then $\mathbb{E}_{D \sim Q(\cdot)}[\bar{\ell}(D)] = 0$. As a quantitative measure, we therefore measure the learning algorithm's *inductive bias towards state* (IB) as

$$\text{IB}(Q) = \mathbb{E}_{D \sim Q(\cdot)}[-\bar{\ell}(D)]. \tag{3}$$

The preceding discussion implies if the learning algorithm respects state, then $\text{IB}(Q) = 0$ for any choice $Q(\cdot)$ over datasets $D \in \mathcal{D}^{\Phi}$. Larger values of $\text{IB}(Q)$ imply that, on average over datasets consistent with the state representation, the learning algorithm returns prediction functions that can be more well-approximated by state.

Second, Definition 2.2 implies that if a learning algorithm respects state, there exists a non-injective function $r(\phi(x))$ such that, for any $D \in \mathcal{D}^{\Phi}$, $\ell^*(r; D) :=$ $\min_{\tilde{s}} \mathbb{E}_{x \sim P(\cdot)}[\ell(s^*(\phi(x); D), \tilde{s}(r(\phi(x))))] = 0$. The best approximating function of state $s^*(\phi(x); D)$ can be compressed and represented in terms of $r(\phi(x))$. Given any representation $r(\cdot)$ of state, we define its reconstruction error as $\epsilon(r, \phi) = \mathbb{E}_{x \sim P(\cdot)}[e(r(\phi(x)), \phi(x))]$ for some chosen reconstruction loss function $e(\cdot, \cdot)$. Therefore if the learning algorithm does not satisfy Definition 2.2, then there exists some representation $r(\cdot)$ of state such that $\epsilon(r, \phi) > 0$ and $\ell^*(r; Q) :=$ $\mathbb{E}_{D \sim Q(\cdot)}[\ell^*(r; D)] = 0$. We therefore measure the learning algorithm's *state recovery* (SR) as

$$\text{SR}(Q) = \min_{r\colon \ell^*(r; Q) = 0} -e(r, \phi). \tag{4}$$

The preceding discussion implies that if the learning algorithm fully reconstructs state, then $\text{SR}(Q) = 0$ for any distribution $Q$ over datasets $D \in \mathcal{D}^{\Phi}$. Larger values of $\text{SR}(Q)$ imply that the best approximation of the learning algorithm based on state uses more of the underlying state $\phi$ i.e., $r(\phi(x)) \approx \phi(x)$.

## 2.2 Implementation via Transfer Learning

In this paper, we study the world model properties of transfer learners. Here, a learner is defined by the model architecture, initialization, and optimization procedure. For example, GPT-2 (Radford et al., 2019) can be a transfer learner with the transformer architecture initialized at GPT-2's weights and optimized with Adam (Kingma & Ba, 2014). The key inputs into calculating our evaluation metrics are: (i) a loss function $\ell(\cdot)$ defined over the outputs; (ii) a reconstruction loss function $e(\cdot)$ defined over the states; (iii) a sampling distribution over inputs $x \sim P(\cdot)$; and (iv) a sampling distribution over datasets consistent with the state representation $D \sim Q(\cdot)$. Given these inputs, we take the following steps.

**Construct synthetic datasets consistent with the state representation.** Given inputs $\{x_1, \ldots, x_n\}$, we sample $x_k$ for $k = 1, \ldots, K$ uniformly at random. For each randomly sampled input $x_k$, we assign an arbitrary output $y_k$ in which we enforce that the outputs are consistent with the underlying state representation. In practice, we consider $y_k \sim \mathcal{N}(0,1)$ and $y_k \sim \text{Bern}(0.5)$. Applying this once produces a dataset $D \in \mathcal{D}^\Phi$, and we repeat this sampling $J$ times to produce datasets $D_1, \ldots, D_J$. This sampling procedure implicitly defines a sampling distribution $Q(\cdot)$ over consistent datasets.

**Apply the learning algorithm on each synthetic dataset.** For each dataset $D_1, \ldots, D_J$, we apply the learning algorithm to produce the models $\widehat{m}(\cdot; D_j)$ for $j = 1, \ldots, J$. We then calculate the associated prediction functions across inputs $x_i$ (sampled from the collection $\{x_1, \ldots, x_n\}$) to produce $\widehat{m}(x_i; D_j)$. This results in $J$ datasets of the form $\{(x_i, \widehat{m}(x_i; D_j)\}$.

**Build multi-task learner to model extrapolations.** Using the datasets from the previous step, we train a multi-task learner that takes as input the true state representation $\phi(x)$ associated with each input and predicts the model extrapolations $\widehat{m}(x; D_j)$ for each context $j = 1, \ldots, J$. By building a representation that's predictive of all $\widehat{m}(x; D_j)$'s, the multi-task learner implicitly maps the state representation $\phi(x)$ to a representation $r(\phi(x))$ that can be used to model all prediction functions simultaneously. In other words, it simultaneously learns $\tilde{s}_j(r(\phi(x)))$ for each context $j = 1, \ldots, J$.

**Calculate inductive bias towards state.** If the original learner has an inductive bias toward state, the multitask learner should be able to predict its extrapolations from the true state. Given the trained multi-task learner, we calculate its average loss on a held-out sample $\widehat{\text{IB}}_j = \frac{1}{m} \sum_{l=1}^{m} \ell(\widehat{m}(x_m; D_j), \tilde{s}_j(r(\phi(x_m))))$ for each context $j = 1, \ldots, J$. We then calculate the inductive bias towards state by averaging across contexts $\widehat{\text{IB}}(Q) = \frac{1}{J} \sum_{j=1}^{J} \widehat{\text{IB}}_j$. To make this estimate more interpretable, in practice, we construct an uninformative benchmark $b$ to predict $\widehat{m}(\cdot; D_j)$ in each context and report $1 - \widehat{\text{IB}}(Q)/\widehat{\text{IB}}_b$ for $\widehat{\text{IB}}_b = \frac{1}{mJ} \sum_{j=1}^{J} \sum_{l=1}^{m} \ell(\widehat{m}(x_m; D_j), b(x_m))$. This normalizes our estimate of the inductive bias towards state into the unit interval, so that values closer to 1 indicate a higher inductive bias towards state. For example, if $\ell(\cdot)$ is squared loss, our benchmark is the baseline variance and the normalized metric is $R^2$.

**Calculate state recovery.** For the multi-task learner, state recovery corresponds to measuring how much the learned $r(\phi(x))$ compresses the state; is the original learner using all of state to extrapolate, or just parts of it? To measure this, we predict the state $\phi(x)$ from the learned representation $r(\phi(x))$; denote the resulting predictions as $\widehat{\phi}$. We then calculate $\widehat{\text{SR}}(Q) = \frac{1}{m} \sum_{l=1}^{m} e(\widehat{\phi}(x), \phi(x))$. As before, we create an uninformative benchmark $c$ and report $1 - \widehat{\text{SR}}(Q)/\widehat{\text{SR}}_c$ for $\widehat{\text{SR}}_c = \frac{1}{m} \sum_{l=1}^{m} e(c(x), \phi(x))$. This again normalizes our estimate of state recovery into the unit interval, so that values closer to 0 imply that the learning algorithm uses more of the underlying state.

This procedure provides two metrics for how much a learner relies on state: inductive bias $\widehat{\text{IB}}$ and state recovery $\widehat{\text{SR}}$. These metrics depend on the implementation of the multitask learner. We consider different multitask learners for the different datasets; for example, for Othello, the multitask learner is a convolutional neural network that's a function of the game board. In practice, these measures may be sensitive to the implementation of the multitask learner. However, we use the same multitask learner for all different models for the same task, ensuring proper comparison. Further, we consider different ablations of the multitask learner in Appendix C — along with a nonparametric approach based on correlation matrices in Appendix G, and reach similar conclusions across methods.

## 3 Orbital Mechanics

Here, we illustrate these metrics on a simple example where learners are applied to data that obey Newtonian mechanics. Specifically, we simulate trajectories of a planet in motion and train a transformer to predict the next location of the planet. We then ask: does fine-tuning a pretrained model to new tasks demonstrate an inductive bias toward the states dictated by Newtonian mechanics? Despite the model performing well on next-token prediction, our metrics reveal a low inductive bias toward state. We demonstrate that the model has recovered piecemeal heuristics rather than a compact world model; while it can recover Newton's law of gravity for narrow slices of the data, it forms other laws for other types of sequences.

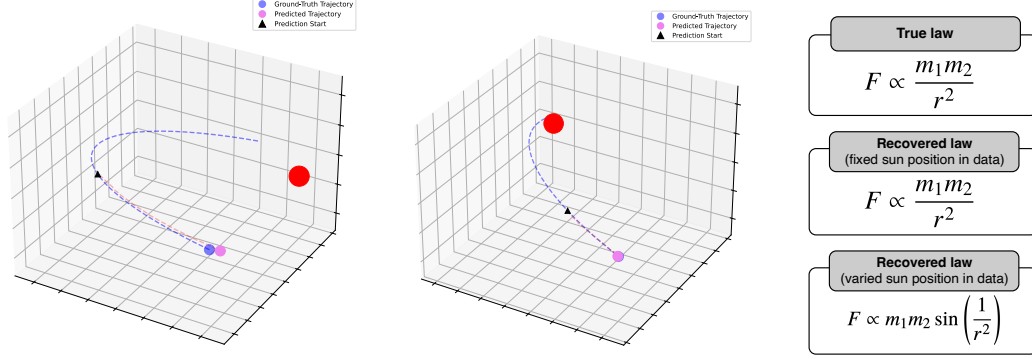

**Figure 1:** In the left and middle, examples of a transformer's generated orbits compared to true orbits. The transformer is given the beginning of an orbit and generates 100 timesteps out. On the right is Newton's law of gravitation along with gravitational laws implied by the model fine-tuned to different slices of force data (given via symbolic regression). While the model exactly recovers Newton's law for narrow slices of fine-tuning data, it struggles for the general dataset.

**Data and pretraining.** We begin by simulating a dataset of sequences where each sequence describes a planet in motion around a sun, i.e. a two-body problem. To do this, we randomly sample initial conditions (e.g. the masses and positions of each planet and their initial relative velocity) and simulate orbits according to Newtonian mechanics. To convert orbits into sequences, we record the $x$ and $y$ coordinates of the lighter planet for 200 time-steps with 5-minute intervals, resulting in sequences of length 400 (e.g. $x_1, y_1, x_2, y_2, \ldots, x_{200}, y_{200}$). We generate 1M such sequences for training (i.e. 400M training tokens) and 10,000 sequences for a held-out set. We train a 12-layer transformer decoder (Vaswani et al., 2017) to predict the next token of each sequence in the training set. We use a modified transformer to model continuous data; see Appendix A for more details.

We evaluate the model's predictions on held-out data. The held-out $R^2$ is above 0.99, indicating very good prediction. The model outperforms a baseline model that always predicts the most recent position, especially as it's forced to generate more of the orbit (Figure 5). The left two panels of Figure 1 shows examples of orbits; given only a few data points for the beginning of the orbit, the model can complete the orbit with high accuracy.

**Is the model a physics learner?** The pretrained model's predictions appear to obey fundamental laws of physics. Here we ask: does the model use laws of physics as an inductive bias when transferring to other problems? To test this, we note that each observation in a sequence of orbits is governed by an 8-dimensional state vector consisting of the masses, relative velocities, and relative positions of each planet. Given the current state of a trajectory, the next position of an orbit is deterministic. If a learner's inductive bias depends on the laws of orbital physics, it must be extrapolating based on state.

We use the metrics described in Section 2 to assess the model's inductive biases. We implement these metrics by simulating small datasets with the same inputs as the pretraining data but different outputs; instead of training the model to predict next-token, we fine-tune it to predict random Gaussian noise on each dataset. The inductive bias (IB) test assesses whether a model's extrapolations across datasets can be predicted by the 8-dimensional state vector; meanwhile, the state recovery (SR) test aims to predict the state vector from the shared representation used by the IB test. We perform the test by generating 5 datasets and fine-tuning the pretrained model on each one to measure its extrapolations. See Appendix B for more details on how we implement these metrics.

We report $R^2$ for both metrics (1 is perfect prediction, 0 is equivalent to a constant baseline). The IB $R^2$ is 0.65, showing that the pretrained model does not have a large inductive bias toward state when it transfers. Meanwhile, the SR $R^2$ is 0.62 for the next-token-pretrained model. This shows that not only do the inductive biases not relate to state, but also that state isn't fully captured.

How can a model perform so well at predicting orbit locations without having inductive biases towards the laws of physics that govern them? We study this question by assessing whether Newton's law of gravitation can be inducted from the model's predictions. Newton's law of gravitation, $F = G\frac{m_1 m_2}{r^2}$,

relates the force $F$ between two objects to their masses $m_1, m_2$ and their squared distance $r^2$. If a learner is transferring based on laws of physics, its extrapolations should obey this law.

We create a sequence-to-sequence dataset where each input is a trajectory and each output is the acceleration magnitude $a$ implied by the state of the orbit, where $a = \frac{F}{m_1} = G\frac{m_2}{r^2}$ (this is equivalent to the gravitational force on a unit-mass object). We then fine-tune the next-token-pretrained model to predict $a$. We then ask: could the model's predicted values of $a$ be used to reconstruct Newton's law of gravitation? To assess this, we perform a symbolic regression (using the *PySR* software (Cranmer, 2023)) of its predicted $a$ values on the true values of $m_2$ and $r$. A symbolic regression is a method to search for a symbolic expression that optimizes a regression-like objective. If the learner has an inductive bias toward Newtonian mechanics, the symbolic regression should recover Newton's law.

We first verify that the symbolic regression indeed recovers Newton's law on real-world data. We then carry out this exercise using the transformer's generations. Rather than recovering Newton's law exactly, the learner recovers piecemeal heuristics. Specifically, when the learner is fine-tuned on only a narrow slice of sequences where the position of the sun is fixed across sequences, the symbolic regression recovers the exact form of Newton's law. However, when we fine-tune on a wider distribution of sequences, where the position of the sun is different for each sequence, it does not; instead, the symbolic regression recovers a nonsensical law of gravity (Figure 1): $F \propto m_1 m_2 \sin\left(\frac{1}{r^2}\right)$. These results demonstrate that rather than building a universal law, the model extrapolates based on piecemeal heuristics; it constructs different laws for different sequences. See Appendix F for further ablations.

## 4 OTHER APPLICATIONS

We now apply our metrics to evaluate the world model properties of learners in other applications. Evaluating world models requires using datasets where ground-truth states are known, and we study two such common types of datasets: lattice problems and the board game Othello.

**Lattice.** One paradigm for assessing world models is studying a model's behavior when it's trained on sequences that arise over lattices (Vafa et al., 2024; Liu et al., 2022). We study a lattice setting similar to the Gridworld example considered in Liu et al. (2022). This setting simulates an agent moving along a line segment with a finite number of positions. Specifically, there is a true state space consisting of $S$ states: $\Phi = \{1, 2, \ldots, S\}$. The language $x$ consists of sequences with three tokens: $\Sigma = \{L, \perp, R\}$. The initial state of the sequence is 1. For a token $\sigma = R$, the state increases by 1, while the state decreases by 1 for $\sigma = L$ and stays the same for $\sigma = \perp$. When the state is 1, the state is at the boundary, so $\sigma = L$ is not a valid token; similarly, when the state is $S$, $\sigma = R$ is not a valid token. All tokens are valid for all other states. The last token of the sequence indicates the final state. We randomly generate sequences over the language by sampling a move uniformly at random over the set of valid moves for each timestep. We initialize the state at 1 and then sample sequences of length 100 over $S = 5$ states. We create a training set of 9.9M tokens and a hold-out set of 44K tokens.

**Othello.** We also study the board game Othello, a common testbed for evaluating the world models of sequence models (Li et al., 2023; Nanda et al., 2023b; Hazineh et al., 2023; Vafa et al., 2024). The game consists of two players taking turns placing tiles on an 8x8 board. Each game of Othello is tokenized into a sequence of at most 60 moves, where each token indicates which of the 60 squares the most recent tile was placed on (the middle four tiles are always occupied). The true state space $\Phi$ corresponds to all 8x8 boards and the mapping $\phi$ converts game sequences into states. Following Li et al. (2023), we consider two different sequence generating processes: **championship**, which corresponds to true gameplay from Othello championships, and **synthetic**, which corresponds to synthetic games generated randomly where each move is sampled uniformly at random from the set of valid moves. We randomly split each dataset into train and hold-out sets. Our training sets contain 7.9M tokens for championship and 60M tokens for synthetic, along with 6K hold-out tokens.

**Models.** We study the world model learning properties for five classes of pretrained sequence models: RNNs (Elman, 1990), LSTMs (Hochreiter, 1997), transformers (Vaswani et al., 2017), Mamba (Gu & Dao, 2023), and Mamba-2 (Dao & Gu, 2024). We use the same number of layers and embedding dimensions for each model so each model has approximately 20M parameters. The only exception is that we find that smaller LSTM and RNN models perform better when trained on lattices, so we use 2-layer models for LSTMs and RNNs for the lattice example. See Appendix A for more information about each type of model.

| | Pretraining | Lattice | | Champ. Othello | | Synthetic Othello | |
|---|---|---|---|---|---|---|---|
| | | IB | SR | IB | SR | IB | SR |
| **RNN** | NTP trained | 0.869 | 1.000 | 0.478 | 0.557 | 0.681 | 0.459 |
| (Elman, 1990) | State trained | 0.980 | 1.000 | 0.507 | 0.572 | 0.483 | 0.464 |
| **LSTM** | NTP trained | 0.850 | 1.000 | 0.792 | 0.520 | 0.840 | 0.443 |
| (Hochreiter, 1997) | State trained | 0.996 | 1.000 | 0.746 | 0.579 | 0.691 | 0.467 |
| **Transformer** | NTP trained | 0.971 | 1.000 | 0.602 | 0.511 | 0.591 | 0.443 |
| (Vaswani et al., 2017) | State trained | 0.925 | 1.000 | 0.734 | 0.628 | 0.706 | 0.451 |
| **Mamba** | NTP trained | 0.695 | 1.000 | 0.552 | 0.550 | 0.465 | 0.408 |
| (Gu & Dao, 2023) | State trained | 0.853 | 1.000 | 0.847 | 0.596 | 0.837 | 0.464 |
| **Mamba-2** | NTP trained | 0.801 | 1.000 | 0.496 | 0.568 | 0.459 | 0.401 |
| (Dao & Gu, 2024) | State trained | 0.840 | 1.000 | 0.693 | 0.582 | 0.736 | 0.447 |

**Table 1:** Inductive bias (IB) and state recovery (SR) metrics (1 is perfect performance, 0 is equivalent to noninformative model). "NTP-trained" represents a model pretrained on next-token prediction, while "state trained" refers to an oracle model pretrained with direct access to state information. While all learners have strong inductive biases and state recovery in the lattice setting, the results are mixed across Othello. While the RNN and LSTM models NTP models are achieving results similar to their state trained capabilities, there is a much larger gap between for the transformer and Mamba models.

For each dataset and model, we consider two types of pretraining objectives. In **next-token prediction** (NTP), we perform the standard pretraining procedure of training a model to predict the next token of each sequence in training data. For example, pretraining applied to Othello would consist of predicting the next move of each game transcript. We also consider an oracle model that's trained to **predict state** (e.g. the true Othello board) (Liu et al., 2022). This oracle model serves as a point of comparison for the next-token prediction model; it helps calibrate the degree to which a model, when given access to ground-truth state, is limited by its architecture. See Appendix A for more information about training and state prediction.

We first demonstrate that all pretrained models perform well at next-token prediction, generating outputs that appear to obey state. Specifically, we measure the fraction of a model's top predictions that are legal in the underlying state, following Toshniwal et al. (2022) and Li et al. (2023). For example, a model's prediction is legal for Othello if the corresponding move is a valid move for the current board implied by the sequence. Table 6 in Appendix E shows the results. All models do very well across all datasets, e.g. every model's top prediction is legal 99% of the time for Synthetic Othello.

### 4.1 INDUCTIVE BIAS METRICS

We now use the metrics described in Section 2 to assess whether these models have inductive bias toward state. Our metrics involve transferring each model to small datasets of randomly generated outputs and assessing how related each model's extrapolation patterns are to the true state. The inductive bias (IB) test aims to predict a model's extrapolations from the true state using a shared representation across datasets, while the state recovery (SR) test aims to predict the original state from the shared representation. Each metric we report is a normalized prediction accuracy so that 0 corresponds to a model with as good predictive performance as a baseline model that makes the same prediction for all inputs and 1 corresponds to perfect prediction (e.g. of state or of extrapolation pattern). For the inductive bias measure, this is held-out $R^2$, and for the state recovery measure, it's 1 minus normalized cross entropy. Appendix B contains more details about how we implement the metrics across datasets and Appendix C contains more ablations.

The results are depicted in Table 1. For the lattice example, almost all inductive biases toward state are close to 1, reaching as high as 0.996 for the LSTM pretrained on state. However, it's not just the models pretrained on state that achieve high inductive bias toward state; the transformer pretrained on next-token prediction has an inductive bias toward state of 0.971. Similarly, all models achieve near perfect state recovery. This shows that not only does the inductive bias of these learners contain information about state; it contains *all* information about state. These results show that our metrics are not unachievable — models can perform well on them.

|  | | **Majority Tiles** | | **Board Balance** | | **Color Parity** | |
|---|---|---|---|---|---|---|---|
|  | Pretraining | NLL ($\downarrow$) | ACC ($\uparrow$) | NLL ($\downarrow$) | ACC ($\uparrow$) | NLL ($\downarrow$) | ACC ($\uparrow$) |
| **RNN** | NTP trained | 0.287 | 0.874 | 0.188 | 0.916 | 0.510 | 0.639 |
|  | State trained | 0.191 | 0.913 | 0.143 | 0.942 | 0.509 | 0.643 |
| **LSTM** | NTP trained | 0.285 | 0.871 | 0.193 | 0.916 | 0.520 | 0.649 |
|  | State trained | 0.160 | 0.938 | 0.131 | 0.946 | 0.523 | 0.654 |
| **Transformer** | NTP trained | 0.237 | 0.894 | 0.174 | 0.926 | 0.510 | 0.654 |
|  | State trained | 0.153 | 0.941 | 0.123 | 0.952 | 0.511 | 0.638 |
| **Mamba** | NTP trained | 0.300 | 0.862 | 0.206 | 0.905 | 0.509 | 0.648 |
|  | State trained | 0.070 | 0.980 | 0.075 | 0.978 | 0.518 | 0.644 |
| **Mamba-2** | NTP trained | 0.274 | 0.879 | 0.184 | 0.914 | 0.515 | 0.637 |
|  | State trained | 0.185 | 0.926 | 0.145 | 0.937 | 0.514 | 0.652 |
| **IB Correlation** | — | 0.637 | 0.651 | 0.617 | 0.643 | 0.695 | 0.347 |

**Table 2:** Results showing transfer performance across new functions of state. "NLL" represents negative log-likelihood (lower is better), and "ACC" represents accuracy (higher is better). "IB Correlation" measures the (unsigned) correlation between each column of results to the inductive bias metrics in Table 1. Transfer learning results are correlated to the inductive bias metrics; models with low inductive bias perform worse at transfer.

While all pretrained models have strong inductive biases in the lattice setting, the results on Othello datasets help differentiate between the transfer abilities of different models. Across datasets, the transformer and LSTM models have the highest inductive bias toward state; however, all inductive biases are lower than they were for the lattice problem. A natural question is how limited each learner is by the model's architecture. By comparing the results for the models pretrained on next token prediction and the state oracle, we see a split: the RNN and LSTM models are achieving performance as good if not better on next-token prediction as they would had they been pretrained on state. Meanwhile, there is a much larger gap between the state oracles and next-token prediction for the transformer and Mamba models. While these models are capable of stronger inductive biases, pretraining on next token prediction does not provide enough guidance to learn state.

The state recovery metrics are low across Championship and Synthetic Othello. These results suggest that the shared representations that are predictive of extrapolations do not carry much state information with them. All models for Synthetic Othello score between 0.40 and 0.50, while the state recovery is somewhat larger for Championship Othello, ranging from 0.51 to 0.63. The discrepancy between Championship and Synthetic Othello is interesting in light of previous findings that models trained on Synthetic Othello are closer to capturing the true world model than those trained on Championship Othello (Li et al., 2023; Vafa et al., 2024). Our results suggest that when we study how these models as transfer learners, the Championship Othello models carry more complete state information.

## 4.2 IMPLICATIONS: TRANSFER LEARNING

The metrics in Table 1 imply that next-token-pretrained models do not have strong inductive biases toward state when trained on Othello. To understand the implications of these results, we study how different models transfer to new functions of state. Specifically, we take the Championship Othello dataset and construct new sequence-to-sequence datasets. The input sequence for each dataset is the original game transcript, and we consider three different output sequences that are functions of state. In "Majority Tiles", each element of the output is 1 or 0 indicating where there are more black or white tiles in the board implied by the sequence so far. In "Board Balance", each element of the output sequence indicates whether black has more pieces in the top half of the board or in the bottom half of the board. Finally, in "Color Parity", the output measures whether the number of black tiles is odd or even. Each of these functions is a deterministic function of state (the board), so learning algorithms that have inductive bias toward state should be better at transfer. We transfer all models for 3000 iterations; see Appendix D for other amounts.

The results are depicted in Table 2. The last row shows the correlation for each metric and the inductive bias measures in Table 1. There is strong correlation across all metrics; models that do better on our inductive bias metrics tend to transfer better to these functions of state. Like Table 2,

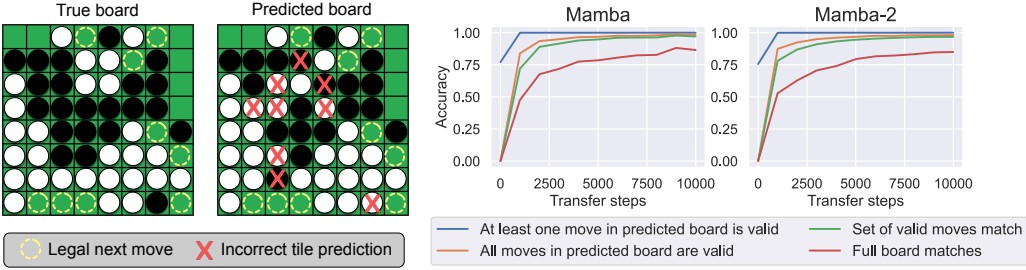

**Figure 2:** On the left, a true Othello board implied by a sequence, and on the right, the predicted board from a model fine-tuned to predict boards. Although the prediction has errors, the set of predicted next tokens exactly matches the true board. On the right, metrics about board reconstruction during fine-tuning. Consistently, even as Mamba models struggle to recover full boards, they recovers them well enough such that the sets of valid next moves match the true boards.

models that are pretrained on state do better than models pretrained on next-token prediction, and the gap is again largest for Mamba. Comparing the transformer and Mamba models, transformers regularly transfer better than Mamba when pretrained on next-token prediction, while the two Mamba models consistently transfer better than transformers when pretrained on (oracle) state information. This shows that while Mamba-like architectures can use state information when it is supplied, the state information extracted by transformers in next-token-prediction pretraining sets them up for transfer learning better than the respective Mamba models. See Appendix D for further analysis.

### 4.3 BUNDLES OF HEURISTICS

The results in this section show that models can perform quite well on pretraining objectives yet have low inductive bias toward state (as measured by our metrics and transfer properties). Here, we try to make sense of this discrepancy. Specifically, we explore the hypothesis that these models are relying on "bundles of heuristics" (Karvonen et al., 2024); functions of the input that lead them to perform well on next-token prediction yet deliver poor inductive biases toward state.

We begin by fine-tuning models pretrained on next-token prediction on Othello to predict the true state (i.e. the board) of each position in the subsequence. Throughout fine-tuning, we reconstruct the fine-tuned model's predicted board for each sequence on held-out data, and record two kinds of metrics. The first is whether the predicted board exactly matches the true board. For the second, we measure how well the set of valid moves in the predicted board matches the set of valid moves in the true board. This is motivated by the fact that even if a predicted board is incorrect, it can still have the same set of valid legal moves. The results are depicted in Figure 2. They point to an intriguing phenomenon: **even when the predicted board is incorrect, the set of legal moves from the predicted board tends to match the set of legal moves from the true board**. These results show that although learners may not carry information about the full board, they carry enough information about the board to perform well at next-token prediction. These findings carry implications for metric design and illustrate a broader principle: models can learn representations that satisfy training objectives without capturing complete world models. This phenomenon parallels observations in LLMs, where models can sometimes answer questions correctly without demonstrating deeper conceptual understanding (Vafa et al., 2024). Future work should investigate similar patterns in domains like physics and navigation.

To further explore this hypothesis, we calculate a variant of our inductive bias metric. Our original inductive bias metric measures how well the extrapolations of a learner can be predicted by a shared function of state (State IB). We calculate another metric which measures how well these extrapolations can be predicted by a shared function of the input sequence. If a shared function of input sequence can predict extrapolations across datasets while a shared function of state cannot, it suggests that a learner is using the same heuristic to guide its extrapolations. We refer to this metric as Heuristic IB and calculate it analogously to State IB (we provide more details in Appendix B).

The results are depicted in Table 3, along with difference in state and heuristic inductive biases (larger implies a larger inductive bias toward heuristics). For almost all models, the heuristic inductive bias is larger than the state inductive bias. For the transformer and Mamba models, the difference

between heuristic and state inductive biases is larger for models pretrained on next-token prediction than models pretrained on oracle state information. The fact that these differences are smaller for state-pretrained models shows that heuristic avoidance is possible for these architectures. However, their next token pretraining encourages them to rely on bundles of heuristics.

## 5 Related Work

One strand of world model research studies whether the outputs of a fixed model accord with a known world model by studying the fixed model's outputs (Vafa et al., 2024). For example, one way that Toshniwal et al. (2022) and Li et al. (2023) study world models is by assessing whether a model trained on sequential game data always predicts legal moves in the underlying game. The question we study is a different yet related question: rather than studying the world model properties of a fixed model, we study what it means to test if a *learner* has a world model embodied in it. This framework could be used, for example, to study how large language models perform in few-shot learning.

Another strand of the literature assesses whether a model's *representations* encode world models without directly studying learning properties. For example, a common method uses probes to assess whether an intermediate representation used by a neural network is predictive of state (Hewitt & Liang, 2019; Li et al., 2021; Abdou et al., 2021; Jin & Rinard, 2023; Li et al., 2023). However, there are open questions about the reliability of probes (Belinkov, 2022), such as appropriate function complexity (Alain & Bengio, 2018; Cao et al., 2021; Li et al., 2023). Our method sidesteps these issues by asking how a model *learns*, rather than what's encoded in its fixed representations.

The methods in this paper are also related to the study of mechanistic interpretability of ML models (Nanda et al., 2023a; Cunningham et al., 2023; Bereska & Gavves, 2024). Closely related to us, Karvonen et al. (2024) find that a GPT model trained on Othello performs internal computations corresponding to "bags of heuristics" rather than a coherent world model. Our procedures differ in aim because 1) we study the world model capabilities of a learner rather than of a fixed model and 2) we do not seek to understand the internal mechanisms governing world model recovery. However, these findings support our analysis of the Othello model relying on heuristics, rather than state, as its inductive bias.

Our examples with orbital mechanics also relates to the large body of work studying AI and physics (Hao et al., 2022; Wu & Tegmark, 2019). The example we study is most closely related to works studying whether AI models can uncover physical laws (Chen et al., 2022; Belyshev et al., 2024). We adapt tools from this literature — such as using symbolic regressions to evaluate AI models — to study the inductive biases of transfer learners (Liu & Tegmark, 2021; Wu & Tegmark, 2019).

This paper studies the problem of evaluating whether the world models of learning algorithms reflect the world models of the real world. There is also a literature on world models in reinforcement learning (RL); while these literatures use similar terms, the goals are distinct. In RL, world models refer to representations (or even the specific neural network) learned by an agent, and their quality is typically measured by how well they perform policy optimization (Ha & Schmidhuber, 2018; Chen et al., 2024). In contrast, the LLM literature focuses on evaluation: a learning algorithm is evaluated by how well it can recover an externally defined mapping from a true world model.

## 6 Conclusion

In this paper, we developed a framework for evaluating whether learning algorithms develop world models by measuring their inductive bias toward state when transferring to new tasks. Our results across applications reveal that while many models perform well on next-token prediction, they can have limited inductive bias toward state and poor transfer properties to new state-based tasks. This suggests that these models may be relying on bundles of heuristics rather than coherent world models.

As described in Section 2, our metrics depend on the implementation of the multitask learner used to model extrapolations and recover state. While we use the same multitask learner for all models within each task to ensure fair comparison, the measures may be sensitive to how this learner is implemented. However, we find similar performance across the different kinds of multitask learners we consider (Appendix C) and for a procedure based on correlation matrices (Appendix G). Further work should prioritize studying the most effective multitask learners for measuring these inductive biases.

**Reproducibility Statement** To ensure reproducibility of our results, we're releasing the codebase used for our experiments. Additionally, all data we create will be made publicly available upon publication. All other datasets used in the paper are already publicly available. All experiments were performed using 1-8 A100 GPUs, ensuring that results are replicable using academic resources.

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

| | | Championship Othello | | | Synthetic Othello | | |
|---|---|---|---|---|---|---|---|
| | Pretraining | State IB | Heuristic IB | Diff. | State IB | Heuristic IB | Diff. |
| **RNN** | NTP trained | 0.478 | 0.926 | 0.448 | 0.681 | 0.857 | 0.176 |
| | State trained | 0.507 | 0.909 | 0.403 | 0.483 | 0.912 | 0.429 |
| **LSTM** | NTP trained | 0.792 | 0.888 | 0.096 | 0.840 | 0.928 | 0.088 |
| | State trained | 0.746 | 0.903 | 0.158 | 0.691 | 0.913 | 0.222 |
| **Transformer** | NTP trained | 0.602 | 0.908 | 0.306 | 0.591 | 0.866 | 0.275 |
| | State trained | 0.734 | 0.838 | 0.105 | 0.706 | 0.806 | 0.101 |
| **Mamba** | NTP trained | 0.552 | 0.748 | 0.197 | 0.465 | 0.814 | 0.350 |
| | State trained | 0.847 | 0.797 | -0.050 | 0.837 | 0.787 | -0.050 |
| **Mamba-2** | NTP trained | 0.496 | 0.784 | 0.288 | 0.459 | 0.848 | 0.390 |
| | State trained | 0.693 | 0.873 | 0.180 | 0.736 | 0.752 | 0.017 |

**Table 3:** We compare the heuristic inductive bias metric (Heuristic IB) to the state inductive bias metric (State IB) proposed in Section 2. The 'Diff' column denotes the difference between state and heuristic inductive biases. Positive values imply dependence on heuristics that do not depend on state.

## A    MODEL AND TRAINING DETAILS

We use the following specifications for each model:

- RNN (Elman, 1990): For Othello, we use an initial 512-dimension embedding layer and pass its output through 8 uni-directional RNN layers with 512 hidden dimensions. For the lattice experiments, the architecture is the same except we use only 2 layers because it optimizes to better in-sample and out-of-sample loss.

- LSTM (Hochreiter, 1997): We use the same specification as for the RNN, except we use 8 LSTM layers instead of RNN layers.

- Transformer (Vaswani et al., 2017): We use a transformer decoder architecture. Following Li et al. (2023), for the non-physics experiments, we consider 8 layers, 8 attention heads, and 512 embedding dimensions. For modeling physics problems, we use a transformer with 12 layers, 16 attention heads, and 512 hidden dimensions. We also modify the transformer so that it can take as input continuous data. Instead of using an embedding lookup table as the first layer, we use a multi-layer perceptron to transform coordinates in Euclidean space to 512-dimensional embeddings.

- Mamba (Gu & Dao, 2023): We first encode inputs with a 512-dimension embedding layer. We then pass inputs through 16 Mamba layers (analogous to 8 layers in a transformer due to how Mamba layers are defined). We use 512 embedding dimensions, 16 for the SSM state expansion factor, 2 for the block expansion factor, and 4 for the convolutional width.

- Mamba-2 (Dao & Gu, 2024): We use the same architecture as for Mamba except the mixer in each block is a Mamba-2 module. We use the same specifications as well: 512 embedding dimensions, an SSM state expansion factor of 16, a block expansion factor of 2, and a convolutional width of 4.

We use Adam (Kingma & Ba, 2014) to optimize each model. We use a learning rate of 6e-4 and decay the learning rate with with 2000 warmup iterations. We use weight decay of $0.1$ and gradient clipping at 1 for each model.

When we pretrain models on next-token prediction, we include a head to predict next tokens (tying its parameter weights to the initial embedding layer parameters). When we fine-tune to predict functions of state, we discard the next-token head and randomly initialize a state head. How we predict state depends on the type of state for each problem:

- Lattice: For the lattice problems, the state corresponds to a categorical variable between 1 and the number of states. We include a state prediction head that forms logits for each state and minimize cross-entropy loss.

- Othello: For Othello (both championship and synthetic), the true state is the board. We represent the board as a 64-dimensional vector (corresponding to an 8x8 grid), where each value takes on one of 3 categorical values (white, black, or unnoccupied). We predict this state using a state prediction head that forms 64x3 logits and minimizing cross-entropy loss summed across all board positions. When we transfer to functions of state, all functions use binary outputs (e.g. 1 if there are more black tiles on the board and 0 otherwise). For these we use a state prediction head forming two logits.

- Physics: For the Newtonian physics problem, the state corresponds to the 8-tuple that includes the position vectors of the two objects, the relative velocity vector of the lighter object, and the masses of the two objects. We predict the state vector normalized across each state dimension and minimize the RMSE loss.

## B    METRIC IMPLEMENTATION DETAILS

**Lattice.** For the lattice example, we create 5 datasets of 500 new examples, $D_1, \ldots, D_5$. For each dataset, we sample sequences uniformly at random among the set of data points and sample outputs from a Bernoulli(0.5) random variable. In our construction we make sure that any two sequences with the same state are mapped to the same output variable. We then fine-tune a model separately for each dataset, resulting in five fine-tuned models $\hat{m}(\cdot; D_1), \ldots, \hat{m}(\cdot; D_5)$. We then calculate the associated prediction functions across all inputs $x_i$ from the original training dataset, resulting in new datasets of the form $\{(x_i, \hat{m}(x_i; D_1))\}, \ldots, \{(x_i, \hat{m}(x_i; D_5))\}$.

For the inductive bias test, we train a model to learn a representation the jointly predicts $(\hat{m}(x_i; D_1), \ldots, \hat{m}(x_i; D_1))$ from the true state $\phi(x_i)$. Since each state is a different categorical variable, the neural network begins with an embedding layer, followed by $L$ feedforward layers with $H$ hidden dimensions and a ReLU nonlinearity (we find the best performance for 3 layers and 64 hidden dimensions). The last layer of the neural network uses a linear transformation to predict the 5 outputs simultaneously. We perform $l_1$ penalization on the penultimate representation to encourage sparsity. We consider $l_1$ penalty values among [0.0, 0.0001, 0.1, 1.0] and choose the penalty with the best validation loss. Since $\hat{m}(x_i; D_j)$ is real-valued (corresponding to the predicted probability of the binary output variable), we train this model for 5000 iterations using a batch size of 600 to minimize the mean-squared error and report the held-out $R^2$.

To perform the state reconstruction test, we predict the original state $\hat{\phi}(x_i)$ from the penultimate representation of the network used for the inductive bias test. We perform this prediction by training a feedforward neural network trained to perform multiclass classification (since each original state is a single class). We find that 2 layers and 512 dimensions does best in practice. We train the model for 5000 iterations with a batch size of 600 to minimize the cross entropy between the predicted and true states. As a baseline, we consider the cross-entropy of a model that always predicts the same value for each class, and report the difference between the baseline model and the model that uses the representation.

**Othello.** Our procedure for Othello follows the same steps as for the lattice example, except we perform adjustments to account for the fact that Othello's state is a 64-dimensional board instead of a single categorical variable. We begin by transferring each model to a dataset of 5 randomly chosen inputs $x_i$ and 5 randomly chosen Binary outputs for each $x$. We perform this transfer exercise for 1000 iterations for 10 seeds, giving us new datasets of the form $\{(x_i, \hat{m}(x_i; D_1))\}, \ldots, \{(x_i, \hat{m}(x_i; D_{10}))\}$.

The implementation of the inductive bias test is the same as for the lattice example except we modify the neural network to account for the fact that our input (state) is an Othello board. Instead of using a simple feedforward network to predict state, we use a convolutional neural network designed to specifically take as input an Othello board. Each Othello board is represented as a 64-dimensional vector $\sigma(x_i)$ where each element is a categorical variable $\{0, 1, 2\}$ indicating whether a black, white, or no tile has been placed on the corresponding square. The first layer of the network begins with an embedding layer, followed by convolutional layers. We follow the convolutional layers with two feedforward layers to predict the output. We again perform $l_1$ penalty on the final layer to encourage sparsity considering the same values as for the lattice example. We find that two convolutional layers, 16 hidden channels, and 64 hidden dimensions for the final feedforward layers performs best. We

|  |  | 1 layer | 2 layers | 4 layers | 8 layers |
|---|---|---|---|---|---|
| **RNN** | NTP trained | 0.484 | 0.504 | 0.486 | 0.478 |
|  | State trained | 0.520 | 0.531 | 0.524 | 0.508 |
| **LSTM** | NTP trained | 0.814 | 0.796 | 0.814 | 0.804 |
|  | State trained | 0.721 | 0.726 | 0.725 | 0.692 |
| **Transformer** | NTP trained | 0.612 | 0.630 | 0.628 | 0.610 |
|  | State trained | 0.722 | 0.725 | 0.717 | 0.711 |
| **Mamba** | NTP trained | 0.533 | 0.522 | 0.538 | 0.532 |
|  | State trained | 0.849 | 0.839 | 0.841 | 0.810 |
| **Mamba-2** | NTP trained | 0.476 | 0.482 | 0.480 | 0.465 |
|  | State trained | 0.673 | 0.677 | 0.671 | 0.635 |

**Table 4:** Ablating the number of layers used for inductive bias prediction.

train this model for 5000 iterations using a batch size of 600 to minimize the mean-squared error and report the held-out $R^2$.

The details for the state reconstruction test in Othello are identical to the lattice example, except instead of predicting a single categorical output we're predicting outputs corresponding to all 64 tiles of the Othello board. We again train the model for 5000 iterations with a batch size of 600 to minimize the cross entropy between the predicted and true states and use the same baseline as for the lattice example.

**Physics.** We follow a procedure analogous to those of the other examples, except we account for the fact that the state corresponds to a vector of continuous, real numbers. We sample 100 random inputs $x_i$ and for each, we sample a Gaussian noise with zero mean and variance of 2.0. We perform the transfer exercise for 100 full-batch iterations to minimize RMSE for 10 seeds, and use the fine-tuned model to generate the new datasets of the form $\{(x_i, \hat{m}(x_i; D_1))\}, \ldots, \{(x_i, \hat{m}(x_i; D_{10}))\}$.

The implementation of the inductive bias and the state reconstruction tests is the same as for the lattice example except we minimize RMSE loss instead of cross-entropy loss.

**Heuristic IB.** The Heuristic IB metric described in Section 4 is analogous to the IB metric for the lattice example, except we predict how well extrapolations can be predicted by a shared representation of the *input sequence* rather than by the state. This is a model from an input sequence (e.g. a sequence of moves in an Othello game) to $K$ extrapolated function values. We train this function by using a transformer to represent the input sequence (using the same configuration described in Appendix A) and including an output head to predict the $K$-vector of extrapolated function values for each input. We optimize parameters to minimize the mean-squared error of the predictions and the extrapolations. We also consider additional architectures to the transformer, such as Mamba and an LSTM, and find similar results; we use the transformer for each bundle of heuristic prediction for consistency.

## C    MULTITASK ABLATIONS

Ablations for different settings for the multitask learner are presented in Table 4 and Table 5.

## D    ADDITIONAL TRANSFER RESULTS

Figure 3 and Figure 4 show examples of training progress for the transfer learning experiments considered in Section 4. These graphs show that the Mamba oracle model has an advantage over the transformer across all stages of fine-tuning on new functions of the Othello state. However, models pretrained on next-token prediction don't achieve this bound. Instead, the transformer trained on next-token prediction transfers better than Mamba trained on next-token prediction despite the superior oracle properties of the Mamba model.

|  |  | 64 units | 128 units | 256 units | 512 units |
|---|---|---|---|---|---|
| **RNN** | NTP trained | 0.504 | 0.514 | 0.534 | 0.555 |
|  | State trained | 0.531 | 0.550 | 0.550 | 0.545 |
| **LSTM** | NTP trained | 0.796 | 0.820 | 0.830 | 0.836 |
|  | State trained | 0.726 | 0.736 | 0.738 | 0.748 |
| **Transformer** | NTP trained | 0.630 | 0.642 | 0.659 | 0.680 |
|  | State trained | 0.725 | 0.737 | 0.746 | 0.750 |
| **Mamba** | NTP trained | 0.522 | 0.549 | 0.567 | 0.580 |
|  | State trained | 0.839 | 0.859 | 0.859 | 0.864 |
| **Mamba-2** | NTP trained | 0.482 | 0.497 | 0.507 | 0.536 |
|  | State trained | 0.677 | 0.690 | 0.702 | 0.711 |

**Table 5:** Ablating the number of hidden units for inductive bias prediction.

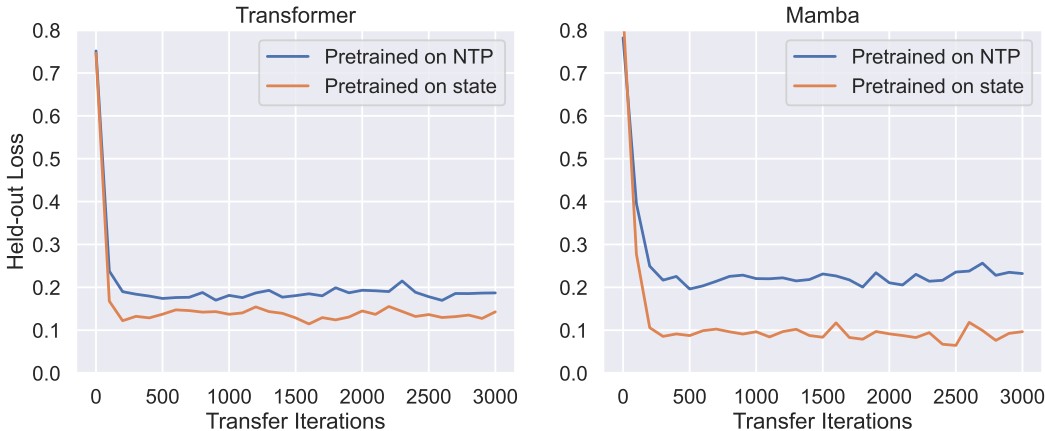

**Figure 3:** Held-out loss progress for transfer learners for the "board balance" transfer task.

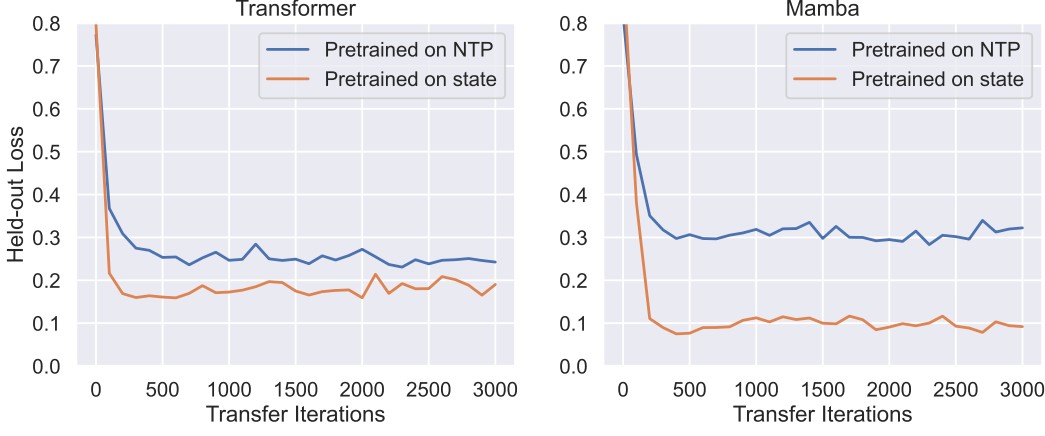

**Figure 4:** Held-out loss progress for transfer learners for the "majority" transfer task.

|  | Lattice | Championship Othello | Synthetic Othello |
|---|---|---|---|
| RNN | 1.00 | 0.905 | 0.995 |
| LSTM | 1.00 | 0.907 | 0.995 |
| Transformer | 1.00 | 0.915 | 0.996 |
| Mamba | 1.00 | 0.890 | 0.996 |
| Mamba-2 | 1.00 | 0.901 | 0.991 |

**Table 6:** Results for the next token test (Toshniwal et al., 2022; Li et al., 2023) for models pretrained on next-token prediction.

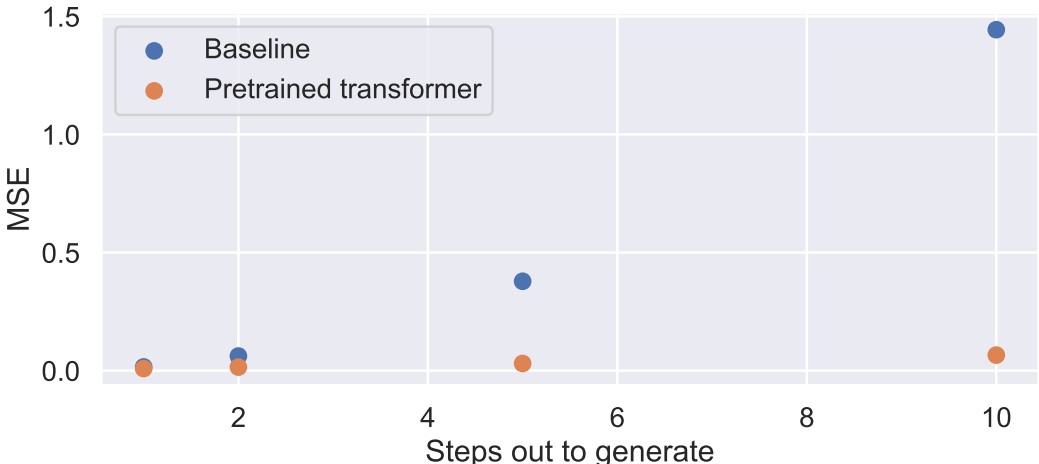

**Figure 5:** MSE for the orbit prediction models. We compare a pretrained transformer to a baseline that always predicts the most recent timestep. The x-axis shows how many tokens the model has to sample.

## E    NEXT TOKEN PERFORMANCE

Table 6 shows results for the next-token test (Toshniwal et al., 2022; Li et al., 2023) for the pretrained models on the lattice and Othello models. It measures the share of top model predictions that are true for the underlying state. All models learn good next token predictions that appear to obey state.

## F    ADDITIONAL SYMBOLIC REGRESSION RESULTS

To more explicitly demonstrate the bundles of heuristics learned by the next-token predictor, we conduct the following experiment: we create five datasets $D_1, \ldots, D_5$, containing 200 random inpputs $x_i$ and the corresponding acceleration magnitude $a_i$ implied by the state of the orbit. We then fine-tune the pretrained next-token-prediction model and the state-prediction model to predict $a$, and generate the extrapolations $\{(x_{i'}, \hat{m}(x_{i'}; D_1)), \ldots (x_{i'}, \hat{m}(x_{i'}; D_5))\}$ on some held-out validation set and use symbolic regression to find the best-fit symbolic equations for the extrapolations.

The state-pretrained model recovered the correct equation, $a \propto \frac{m_2}{r^2}$ in the majority of the five seeds, whereas the next-token predictor recovers five different equations for the five seeds, as shown in Equation (5) - Equation (9).

$$a \propto e^{-0.11r} m_2 \tag{5}$$

$$a \propto \frac{m_2}{r^2} \tag{6}$$

$$a \propto e^{-0.08r}(m_2 + 0.13) \tag{7}$$

$$a \propto e^{-0.11r + \cos m_2} m_2 \tag{8}$$

$$a \propto e^{-0.10r} \sin m_2 \tag{9}$$

Since each subsample recovers a different physical law, this provides futher evidence for the notion that the model constructs different, piecemeal laws for different datasets of sequences.

## G    Correlation-based Metrics

Here we implement an additional procedure for estimating the inductive bias and partial reconstruction metrics from Section 2.2. This procedure begins by following the basic setup in Section 2.2: we construct synthetic datasets that obey the state representation, and apply the learning algorithm. However, instead of estimating inductive bias and state recovery with a multitask learner, we estimate them nonparametrically.

Specifically, after applying the learning algorithm across the $J$ datasets, we collect the extrapolations $\{x_i, \hat{m}(x_i, D_j)\}$ for some set of held-out points $x_1, \ldots, x_n$ that are shared across datasets. The inputs $x_1, \ldots, x_n$ should satisfy the following two properties:

1. There exists an $i \neq j$ such that $x_i$ and $x_j$ have the same state, i.e. $\phi(x_i) = \phi(x_j)$.
2. There exists an $i' \neq j'$ such that $x_{i'}$ and $x_{j'}$ have different states, i.e. $\phi(x_{i'}) \neq \phi(x_{j'})$.

We then construct the $n \times n$ correlation matrix $\Sigma$ such that $\Sigma_{i,j} = \text{corr}(\hat{m}(x_i, \mathbf{D}), \hat{m}(x_j, \mathbf{D}))$, where $\hat{m}(x_i, \mathbf{D}) = (\hat{m}(x_i, D_1), \ldots, \hat{m}(x_i, D_J))$ is the vector of extrapolations across datasets. Intuitively, $\Sigma_{i,j}$ describes how similar the extrapolations are for datapoints $x_i$ and $x_j$; how predictable is $x_j$'s extrapolation from that of $x_i$'s?

If a learner respects state, its extrapolations for two points in the same state will be perfectly correlated. Similarly, if a learner is fully reconstructing state, it will have zero correlation for pairs of points that are not in the same state. Therefore, to estimate inductive bias, we compute the average correlation between points that have the same state:

$$\mathbb{E}[\Sigma_{i,j} | \phi(x_i) = \phi(x_j), i \neq j]. \tag{10}$$

A value of 1 implies perfect inductive bias toward state. Similarly, estimating state recovery involves computing the average absolute correlation between points that don't have the same state:

$$\mathbb{E}[|\Sigma_{i,j}| | \phi(x_i) \neq \phi(x_j)]. \tag{11}$$

Nonzero values mean that a learner is extrapolating based on only partial functions of state. Our definition of state recovery in Section 2 would involve negating Equation (11) so that higher values are better. However, because Equation (10) and Equation (11) are directly comparable, it is easier to compare them without negating Equation (11). Instead we report Equation (11) directly and refer to it as **state coarseness** (SC).

Because correlation-based measures of inductive bias and state coarseness are directly comparable, we additionally report the ratio of the two values: IB/SC. The ratio summarizes how much more correlated extrapolations of same-state pairs are than different-state pairs. Larger values mean larger levels of same-state correlation relative to different-state.

Below we perform the main analyses in Section 3 and Section 4 with the new correlation-based metrics, finding similar results across methods.

### G.1    Physics.

For the physics problem, we create 25 datasets of 100 data points, whose outputs are random Bernoulli draws that are constructed to be consistent with the discretized state-space, where each continuous state is mapped first to one of ten bins based on the magnitude of its norm. We train each learner for 100 iterations then estimate the correlation matrix using 100 held-out sequences. We repeat each experiment 4 times with different random seeds to estimate standard errors. We report the inductive bias, state coarseness, and the correlation ratio in Table 7.

### G.2    Lattice and Othello.

For the lattice problem, we create 25 datasets of 100 data points, whose outputs are random Bernoulli draws that are constructed to obey state structure. We train each learner for 300 iterations. We then

| Training | Inductive Bias | State Coarseness | Ratio |
|---|---|---|---|
| NTP trained | 0.237 (0.041) | 0.227 (0.040) | 1.045 (0.005) |
| State trained | 0.307 (0.042) | 0.264 (0.039) | 1.171 (0.018) |

**Table 7:** Comparison of inductive bias (same-state correlation) and state coarseness (different-state correlation) measures for the transformer model under next-token prediction (NTP) and state-based training, along with their ratios, for the **physics** problem. Larger ratios mean that the extrapolations of a learner have larger correlation among data points in the same state than among points in different states. Similar to the results in Section 3, the NTP-learner extrapolates using less of the world model than the state-trained learner. Standard errors are in parentheses.

| | | Lattice | Champ. Othello | Synthetic Othello |
|---|---|---|---|---|
| **RNN** | NTP trained | 1.967 (0.046) | 1.234 (0.047) | 1.063 (0.013) |
| | State trained | 2.435 (0.088) | 1.139 (0.011) | 1.107 (0.028) |
| **LSTM** | NTP trained | 2.758 (0.029) | 1.061 (0.013) | 1.037 (0.005) |
| | State trained | 2.875 (0.159) | 1.046 (0.006) | 1.021 (0.003) |
| **Transformer** | NTP trained | 3.592 (0.010) | 1.324 (0.029) | 1.487 (0.051) |
| | State trained | 5.428 (0.357) | 1.593 (0.084) | 1.366 (0.029) |
| **Mamba** | NTP trained | 2.847 (0.080) | 1.361 (0.027) | 1.486 (0.044) |
| | State trained | 3.183 (0.034) | 1.515 (0.044) | 1.345 (0.063) |

**Table 8:** The **ratio** between the correlation-based measures of inductive bias and state coarseness. Larger means that the extrapolations of a learner have larger correlation among data points in the same state than among points in different states. Ratios are again correlated to the transfer learning performance for Othello (Section 4); the correlations are 0.712, 0.643, and 0.526 for board balance, majority tiles, and color parity, respectively. Standard errors are in parentheses.

estimate the correlation matrix using 100 held-out sequences. For the Othello datasets, we follow a similar procedure, training for 100 iterations across 25 datasets of 100 data points each. For Othello, randomly drawn sequences will have very few sequences with the same state. Because of the requirement that there be sequences with the same state, we sample held-out sequences in a way that includes more sequence per state. Specifically, we create a dataset of 84 length-8 game beginnings that contain a lot of state overlap. We repeat each experiment 4 times to estimate standard errors.

We report the correlation ratio (IB/SC) as our main summary, which measures how much stronger the inductive bias toward state is than the state coarseness. The results are depicted in Table 9. The trends are similar to those in Section 2: models do well on lattice (reaching ratios as high as 5.4 for the state trained transformer), but considerably worse on Othello (the average correlation for same-state pairs is never more than even twice as high as the average correlation for different-state pairs). We note that the exact numbers don't match the metrics in Section 4 due to scaling differences, i.e. correlation can be between -1 and 1 while the metrics in Section 4 are all normalized to be between 0 and 1. However the orderings are similar, and the ratios are again correlated to the transfer learning performance for Othello; the correlations are 0.712 for board balance, 0.643 for majority tiles, and 0.526 for color parity. We include individual results for correlation-based inductive bias and correlation-based state coarseness in Tables Table 9 and Table 10.

|  |  | Lattice | Champ. Othello | Synthetic Othello |
|---|---|---|---|---|
| **RNN** | NTP trained | 0.683 (0.015) | 0.778 (0.012) | 0.965 (0.005) |
|  | State trained | 0.786 (0.016) | 0.868 (0.009) | 0.913 (0.013) |
| **LSTM** | NTP trained | 0.757 (0.011) | 0.947 (0.011) | 0.975 (0.002) |
|  | State trained | 0.824 (0.017) | 0.963 (0.006) | 0.983 (0.002) |
| **Transformer** | NTP trained | 0.744 (0.016) | 0.784 (0.017) | 0.878 (0.008) |
|  | State trained | 0.957 (0.003) | 0.800 (0.016) | 0.859 (0.012) |
| **Mamba** | NTP trained | 0.706 (0.024) | 0.669 (0.018) | 0.876 (0.009) |
|  | State trained | 0.730 (0.011) | 0.793 (0.011) | 0.827 (0.013) |

**Table 9:** The correlation-based measure of **inductive bias**. Large values means that the extrapolations of a learner are correlated among data points in the same state. Most models have high correlation-based inductive bias, and the state trained models are consistently larger than the NTP-trained ones. Standard errors are in parentheses.

|  |  | Lattice | Champ. Othello | Synthetic Othello |
|---|---|---|---|---|
| **RNN** | NTP trained | 0.348 (0.013) | 0.635 (0.030) | 0.908 (0.015) |
|  | State trained | 0.324 (0.006) | 0.763 (0.014) | 0.827 (0.026) |
| **LSTM** | NTP trained | 0.275 (0.006) | 0.894 (0.020) | 0.941 (0.007) |
|  | State trained | 0.289 (0.013) | 0.922 (0.011) | 0.963 (0.004) |
| **Transformer** | NTP trained | 0.209 (0.010) | 0.594 (0.025) | 0.593 (0.017) |
|  | State trained | 0.179 (0.012) | 0.507 (0.023) | 0.630 (0.016) |
| **Mamba** | NTP trained | 0.249 (0.012) | 0.493 (0.021) | 0.592 (0.021) |
|  | State trained | 0.229 (0.005) | 0.526 (0.021) | 0.621 (0.035) |

**Table 10:** The correlation-based measure of **state coarseness**. Large values means that extrapolations of a learner are correlated among data points in *different states*. While these correlations are low for lattice, they're high for Othello domains, especially for the RNN and LSTM models. Paired with Table 9, these results show that while learners are indeed extrapolating based on state, they are very coarse functions of state. Standard errors are in parentheses.