# OpenReview forum: "Evaluating the World Models Used by Pretrained Learners"
_ICLR.cc/2025/Conference — Submitted to ICLR 2025_

### Official Review · Reviewer_PEo1 · 2024-11-02

**Soundness:** 2
**Presentation:** 2
**Contribution:** 2
**Rating:** 5
**Confidence:** 4

**Summary:**

The paper presents new methods to study whether a trained model’s performance is made possible by an internal “world model”, meaning representations of the underlying “state” which generates the data. For example, in their gravitational trajectories example, the data itself is the observed position of the planets at different time-steps, while the underlying state includes their velocities and masses (which determine future positions). To that end, the authors present and motivate two metrics (IB and SR) meant to capture the extent to which a learned model’s behavior can be understood as depending only on the (unseen) states, and as depending on all such states (instead of a coarser classification). Experiments are run on several different sequential datasets with a known underlying state, testing how different architectures score on these metrics against random baselines, as well as a baseline directly trained on state observations. In some cases, other methods are used to interpret model behavior, like regression. The authors indeed find that different architecture and task pairs score differently on these metrics, and discuss their interpretation.

**Strengths:**

The topic is pertinent and of broad scientific interest and the work is ambitious. They propose a set of concepts and formal metrics that (to my knowledge) are novel and are a potentially valuable contribution. These metrics are connected to empirical experiments in a potentially fruitful way. The baselines seem reasonable.

The discussion of related work seems adequate (although it would be hard for me to tell whether it’s missing something important). The orbital mechanics domain is elegant and seems like a good toy domain for future work. The other domains have appeared in previous related work, and so seem like a reasonable choice. The empirical observation in Section 4.3 that the learned model fine-tuned on state prediction is correct about legal moves but not the whole board is interesting.

**Weaknesses:**

Unfortunately, I believe there are several important weaknesses with the paper’s main contributions (which might or might not be easy to address), as well as its presentation.

The main contribution of the paper are the metrics defined in Section 2.1. While their motivation is understandable, it is not immediately clear, nor in my opinion sufficiently justified, why these metrics are tracking what the authors propose they are tracking (i.e. the extent to which a model’s internal mechanisms correspond to the world-model generating the data). While the binary definitions (2.1 and 2.2) make intuitive sense (modulo some presentation problems which I discuss below), there’s no explanation for why their quantitative extension is appropriate for the author’s purposes. For example, even when dividing by the random baseline, it is hard to interpret what different scales correspond to. For example, whether a 0.6 IB represents, intuitively speaking, a model clearly using a pretty accurate state-model (albeit still short of a perfect one), or instead a 0.6 IB can be attained “quite easily” even when the model is just using a bunch of heuristics, and very far away from having a coherent world model. Of course, I understand the results in Tables 1 and 2 are interpreted as evidence that these metrics are tracking the desired concepts. But I do think deeper discussion is required on why this intuition is warranted, and what these metrics in particular add to the picture. For example, on the gravitational orbits experiments, the authors also turn to symbolic regression to elucidate whether the model can or cannot be understood as comprehensively applying the generating laws, and this seems like a much more straightforwardly interpretable result.

On a similar note, my understanding is that the calculated metrics only provide potentially interesting information when the states used to compute them are the real states used to generate the data. This of course reduces the direct applicability of this technique in cases of particular importance, like the LLM-related examples which the authors present as motivation. The technique might still be useful in several contexts or more indirect ways, but I’d like to see more explicit discussion of them -- otherwise the reader might be led to believe they don’t exist.

The settings the authors choose for their experiments make sense, but my understanding is that they are not necessarily an important contribution: we don’t really care that much whether a specific architecture, run on a setting with a specific encoding, does or doesn’t recover a particular functional form. Rather, they serve mostly as exemplifications and explorations of the use of the proposed metrics, which are indeed the main contribution (and are ultimately aimed at applications to practical models of the kind referred to in the opening paragraph).

The observation in Section 4.3 that models reconstruct the board well enough to match legal moves, but not perfectly, is very interesting, and hints towards some frugal, partial construction of a world-model shaped by the immediate training incentives. Especially since this is so interesting, it would have been useful to include in the main text a more detailed discussion on what it could mean, as well as possibly point to empirical follow-ups in settings other than Othello. (Such an addition is not possible at this stage).
Finally, I also have some major reservations about mathematical presentation and writing, which I present below.

**Questions:**

I start with some other major conceptual considerations:

The mathematical presentation in Section 2 seems convoluted and unclear.
If my understanding is correct, it can be easily proven that Definition 2.1 is equivalent to a very simple mathematical property: m decomposing through /phi (in the support of P). That is, the existence of a function f such that m = f o /phi. Stating this explicitly as the definition would be easier for the reader, as opposed to going through s*. On a similar note, the name “inductive bias towards state” seems like a strange one for this property, especially because it sounds quantitative (L could have more or less inductive bias) when the definition is in fact binary. It would seem better to call the definition “L learns functions of state” (or something like that), and reserve discussions of inductive bias for the latter quantitative versions.

If my understanding is correct, it can be easily proven that Definition 2.2 is equivalent to another very simple mathematical property: the fact that the function f from the previous definition is not injective. This again is easier to understand. Here again the name doesn’t seem as representative as it could be. To me, it would seem more intuitive to name the positive property (f being injective) as “L fully uses state” (or “fully separates states”, or something like that), and then refer to the negation as “L doesn’t fully use state”.
In fact, these names seem not only like suboptimal choices, but also possibly misleading. Consider, for example, a learning algorithm L that, for any dataset D, creates a model m which maps every input to a single constant output. This would satisfy Definition 2.1. Yet it seems misguided to say that this useless L has “inductive bias towards state”. I think similarly counterintuitive edge cases exist for Definition 2.2.

I understand you might want to introduce other terms, like s*, to make your quantitative definitions in Section 2.1 possible. But in any event this should come in Section 2.1, after the simple binary definitions are transparently expressed. Although I also suspect there could be cleaner ways to phrase them that become obvious once you rewrite Definitions 2.1 and 2.2 as above, or even better quantitative definitions altogether.

On a different important note, in several places the authors mention that this paper studies whether a “learner” has a world-model, which sets the work apart from more extensive existing work on whether a fixed model has learned a world-model. Nowhere in the text is it particularly clear what they mean by this, but my interpretation is that their metrics are defined for learning algorithms L (plus distributions over data Q, etc.), as opposed to a single learned model m. If this is indeed the motive for their words, then I’m not sure this really grants a substantive distinction, or sets the work apart from other work in an important way. The reason is that the interesting parts of their definitions all pertain to a single model m, trained on a single dataset D. And then, they simply average in a certain way over datasets D to come up with a metric for the whole learning algorithm L. While this is a natural step, it doesn’t seem like a substantive contribution, and I would have appreciated more thorough discussion of how this new level can open new possibilities for the study of world-model construction.

On another note, Section 3 mentions that (for the models trained on the general dataset as opposed to the narrow slices) the law recovered through regression is not the generating one, and this implies that “the model extrapolates based on piecemeal heuristics; it constructs different laws for different sequences”. This implication doesn’t seem justified enough in the main text. Most notably, how do the authors know that the reason regression finds the wrong law is that “the model constructs different laws for different sequences”, as opposed to something like, “the model constructs the same law for all sequences, but it is simply the wrong law”? The fact that training on narrow slices does recover the correct law doesn’t seem to imply the former over the latter. I can imagine that through experimentation, or also through observing the next-token performance of these models, the authors have developed robust intuitions about which of the two is happening. But if so they are not sufficiently transmitted in the main text.

Now a less important conceptual consideration:
Given your observation that models trained on the general dataset for Orbital Mechanics don’t recover the correct law, it would be positive to include some discussion of the extent to which such architectures can computationally reproduce such laws. That is, the extent to which this failure is due to inductive training bias, or to fundamental limitations of the computational architecture. I understand state-trained models provide some (possibly conclusive) evidence in the direction of “the model architecture can perfectly compute the correct law in a forward pass if the learned model is the correct one”. Or maybe it is obvious that the architecture can implement this computation, or some existing literature has studied this in-depth. But I would have benefitted from making this explicit.

Finally, some stylistic points:
In a few places (mostly the introduction), the authors mention they study world-modelling capabilities along two distinct dimensions: “inductive bias towards state” and “recovering of state”. Of course, these terms refer to Definitions 2.1 and 2.2, but their intuitive meaning was not at all clear to me from reading just their short descriptions in the introduction (and not even immediately upon reading the Definitions, as I’ve explained above), and again I believe they could be made immediately precise by just using the right words: whether the learned model decomposes through states, and whether it makes use of all state distinctions.

In a few places the authors mention new functions (for example, “how much do new functions depend solely on state versus non-state functions of input?”), but to my understanding these “new functions” are simply “the learned model”, and should better be called that.
It seems unnecessary to introduce the variable L, since m\hat ( · ; · ) can already be seen as the learning algorithm, which provides a learned model m\hat( · ; D) for every dataset D, which provides an output m\hat(x ; D) for every input x.
(If for some reason the authors don’t want to incorporate this change, then at least they should introduce the variable L earlier, when the term “learning algorithm” is introduced.)

The variable you introduce for the state space is /Phi, but you later use \mathcal{Q}, for example in equation 1 and Definition 2.2.
Definitions 2.1 and 2.2 depend on /phi, and this should be made explicit in their names (for example, “inductive bias towards state /phi”).
The quantitative metrics should be definitions, since they are more central to the paper (or non-trivial) than the binary definitions 2.1 and 2.2.

A large value of IB(Q), that is, a high “inductive bias towards state”, indicates that the learned model cannot be well-approximated as decomposing through state. This is intuitively backwards: a high “inductive bias towards state” intuitively corresponds to the model being likely to behave in terms of those states, and not the other way around.

I understand that the function /phi is supposed to be the “real” state function throughout, meaning the one actually used to generate the data. This is especially obvious in the experimental sections, but should somewhere be mentioned explicitly, especially because otherwise the definitions in Section 2 don’t necessarily have the same intuitive meaning or motivation that you discuss.
In a couple places, you introduce the Orbital mechanics setting with the sentence “The true world model is the world”, meaning, of course, that the data is generated from Newton’s law, and this is a physical law that approximately describes the real world. But I find this sentence doesn’t add anything, adds unnecessary pomp, and in fact slightly misleads: my first reaction to such a sentence (especially given your motivating examples with GPT-2) was that you would have found a way to extract world models from “messy, complex, real-world data” (like GPT-2’s training corpus). But that is not the case, and your use of Orbital mechanics as an experimental setting is not necessarily different or more interesting than other latent-function-approximation tasks.

Probably the footnote of Figure 1 should quickly mention how these laws are extracted from model behavior, which is, of course, through symbolic regression.
Finally, there are numerous infelicities or mistakes in the text, and I present here just a few representative ones:
- First sentence in the introduction is awkwardly phrased
- End of page 1: “studied by studying”.
- The introduction seems too short to make some of the ideas clear (see especially my comment above about the descriptions of “inductive bias towards state” and “recovering state”).
- When defining what it means for a dataset D to be consistent, you say “with_then_”, which should instead be either “if_then_” or “with_we have_”.
- Introducing /epsilon is probably not necessary.
- Right before Section 2.1, you say “s* has an inductive bias towards state”, which should be “L” instead.
- Before equation 3: “for any chosen distribution over Q over dataset D”.
- “Build multi-task learner to model extrapolations” -- typo in first sentence of this paragraph
- “A symbolic regression is a method…” typo in this sentence”
- “piece-meal” -- inconsistent use of hyphen
- In the reproducibility statement: “All other datasets used in the paper already publicly available.”

---

> ### Author Response · Authors · 2024-11-21
> **Author Rebuttal (1/4)**
>
> We greatly appreciate your detailed read of the paper and your thorough review. We’re glad that you found our work to be novel and of broad scientific interest. Your comments were incredibly constructive and we believe we've addressed all of them in the updated paper -- addressing your comments has improved the clarity of the paper and sharpened the main conceptual points.
>
> To summarize, we have:
> - Revised the framework section in response to your comments.
> - Included additional ways to estimate our metrics that reach the same conclusions (Appendix G).
> - Added additional empirical results (e.g. more symbolic regression experiments in Physics in Appendix F)
> - Sharpened the writing overall.
>
> We have more details below.
>
> > _The main contribution of the paper are the metrics defined in Section 2.1. While their motivation is understandable, it is not immediately clear, nor in my opinion sufficiently justified, why these metrics are tracking what the authors propose they are tracking... For example, even when dividing by the random baseline, it is hard to interpret what different scales correspond to... On the gravitational orbits experiments, the authors also turn to symbolic regression... and this seems like a much more straightforwardly interpretable result._
>
> You bring up a good point about scaling, e.g. interpreting whether IB scores of 0.6 are small or large. As you mention in your review, this is one of the reasons we include baselines such as a model trained on state. Nevertheless, in order to provide more clarity, we've included new quantitative measures in Appendix G that correspond to correlations, where the scaling is more intuitive. At a high level, these additional metrics measure whether a learner's extrapolation behavior for two data points in the same state is more correlated than its extrapolations in different states. We've re-implemented the analyses in our submission with these new metrics and find similar results (we've re-implemented the metrics for all models except for Mamba-2, which we did not have time to re-run during the rebuttal period). We include more details and discussion in our updated appendix.
>
> We're also glad you appreciated the interpretability of the symbolic regression example. While symbolic regression is suitable for physics, where the underlying relationships tend to be mathematically structured and well-defined, the nature of the state space structure for other domains like Othello make other analysis methods more suitable.
>
> > _My understanding is that the calculated metrics only provide potentially interesting information when the states used to compute them are the real states used to generate the data... The settings the authors choose for their experiments make sense, but... we don’t really care that much whether a specific architecture, run on a setting with a specific encoding, does or doesn’t recover a particular functional form. Rather, they serve mostly as exemplifications and explorations of the use of the proposed metrics, which are indeed the main contribution_
>
> You bring up an important point: our metrics are intended for domains where there is some notion of true state used to generate data. In doing this we've followed much of the prior literature on assessing the world models of LLMs, which focus on synthetic domains such as games [1, 2], logic puzzles [3], and navigation [4]. These testbeds are important for developing general metrics and assessing model capabilities; e.g. what can be said about a transformer's world model if it can't extrapolate in even simple domains? Settings where state is known also come up in more complex systems (e.g. anything that obeys a DFA), such as search engines, control systems, and genetics [5].
>
> We agree that the specific experimental findings are not the main contribution. Rather, the experiments serve two key purposes: (1) They demonstrate how to implement our metrics in practice and validate that they capture meaningful properties of learners. (2) They reveal interesting phenomena about how different architectures learn from data - e.g. that learners pretrained on next-token prediction can recover enough of state to make accurate next token predictions despite incoherent full states (Figure 2). This also follows the LLM world model literature [1, 2, 3, 4]. These insights, while derived from synthetic settings, help us understand potential limitations of current approaches.
>
> [1] Toshniwal, Shubham, et al. “Chess as a Testbed for Language Model State Tracking.” AAAI. 2022
> [2] Li, Kenneth, et al. “Emergent World Representations: Exploring a Sequence Model Trained on a Synthetic Task.” ICLR. 2023
> [3] Li, Belinda Z., et al. “Implicit Representations of Meaning in Neural Language Models.” ACL. 2021
> [4] Vafa, Keyon, et al. “Evaluating the World Model Implicit in a Generative Model.” NeurIPS. 2024
> [5] Gribkof, Eric. “Applications of Deterministic Finite Automata.” Lecture note, UC Davis EC120, Sprint 2013

---

> > ### Author Response · Authors · 2024-11-21
> > **Author Rebuttal (2/4)**
> >
> > > _The observation in Section 4.3 that models reconstruct the board well enough to match legal moves, but not perfectly, is very interesting, and hints towards some frugal, partial construction of a world-model shaped by the immediate training incentives. Especially since this is so interesting, it would have been useful to include in the main text a more detailed discussion on what it could mean, as well as possibly point to empirical follow-ups in settings other than Othello. (Such an addition is not possible at this stage)._
> >
> > Thank you for the comments on the board reconstruction example (Sec 4.3); we're glad you found it interesting. These findings show how it's possible for models to learn representations that satisfy training objectives without capturing complete world models. We've updated the discussion in our paper with the following points:
> > - Implications for metric design (measuring whether models learn complete representations versus minimal ones that satisfy narrow objectives)
> > - Connecting to phenomena from LLMs (LLMs answering questions correctly without understanding concepts)
> > - Future empirical follow-up: description of how this could be assessed in other domains like physics and navigation.
> >
> > > _If my understanding is correct, it can be easily proven that Definition 2.1 is equivalent to a very simple mathematical property: m decomposing through /phi (in the support of P)... Stating this explicitly as the definition would be easier for the reader, as opposed to going through s*... It would seem better to call the definition “L learns functions of state” (or something like that), and reserve discussions of inductive bias for the latter quantitative versions._
> >
> > Thank you for this suggestion. In the revised version, we only introduce s* to motivate our quantitative evaluation metrics for inductive bias towards state and state recovery. We now state Definition 2.1 as requiring the existence of some function f (which may depend on the dataset D) such that m(x; D) = f(\phi(x); D). We have also relabeled Definition 2.1 as capturing whether the learning algorithm “respects state,” which both better captures its binary nature and its interpretation.
> >
> > > _If my understanding is correct, it can be easily proven that Definition 2.2 is equivalent to another very simple mathematical property: the fact that the function f from the previous definition is not injective... Consider, for example, a learning algorithm L that, for any dataset D, creates a model m which maps every input to a single constant output. This would satisfy Definition 2.1. Yet it seems misguided to say that this useless L has “inductive bias towards state”._
> >
> > Thank you for this excellent point. We introduce Definition 2.2 precisely to deal with cases in which a learning algorithm respects state (i.e., always returns a constant prediction function) but does not learn non-trivial functions. Towards this, we have restated Definition 2.2 as a refinement of Definition 2.1 (respecting state) and simplified its statement to be about the injectiveness of the representation of state.
> >
> > We now introduce s* to define our quantitative evaluation metrics for “respect” of state and “fully reconstructs” state.
> >
> > > _I understand you might want to introduce other terms, like s*, to make your quantitative definitions in Section 2.1 possible. But in any event this should come in Section 2.1, after the simple binary definitions are transparently expressed._
> >
> > Thank you for your suggestion. We agree that s* is not needed until we introduce the quantitative evaluation metrics introduced in Section 2.1. We have implemented this change in the revised draft.

---

> > > ### Author Response · Authors · 2024-11-21
> > > **Author Rebuttal (3/4)**
> > >
> > > > _In several places the authors mention that this paper studies whether a “learner” has a world-model, which sets the work apart from more extensive existing work on whether a fixed model has learned a world-model... The interesting parts of their definitions all pertain to a single model m, trained on a single dataset D. And then, they simply average in a certain way over datasets D to come up with a metric for the whole learning algorithm L._
> > >
> > > This is a good question. To your earlier comment, the definitions we propose are “binary” and we then provide evaluation metrics to measure how close or how far a learning algorithm is to satisfying those definitions. The first definition of respecting state is about how a particular function behaves across inputs that are mapped to the same state. IB measures whether the learning algorithm always returns a function satisfying that property and so it indeed averages over datasets. However, state recovery does not, because by definition it measures properties that hold _across_ learned models.
> > >
> > > Of course there are other ways to evaluate whether a learning algorithm satisfies these definitions. In Appendix G, we now describe a correlational based version. If the correlation between all points with the same state is 1 and correlation between points with different states is 0, then respecting and fully reconstructing state are satisfied, respectively. What's reassuring is that these alternative ways of evaluating these definitions all point in the same direction.
> > >
> > > > _How do the authors know that the reason regression finds the wrong law is that “the model constructs different laws for different sequences”, as opposed to something like, “the model constructs the same law for all sequences, but it is simply the wrong law”?_
> > >
> > > This is a good question. We've added an experiment where we repeatedly subsample different slices of the training data and fit symbolic regressions to each slice. Each subsample recovers a different physical law. This experiment is evidence for the former argument: that the model constructs different laws for different sequences. You can see more information in Appendix F.

---

> > > > ### Author Response · Authors · 2024-11-21
> > > > **Author Rebuttal (4/4)**
> > > >
> > > > > _Given your observation that models trained on the general dataset for Orbital Mechanics don’t recover the correct law, it would be positive to include some discussion of the extent to which such architectures can computationally reproduce such laws._
> > > >
> > > > This is a great point. As your review acknowledges, our state-trained model recovers the correct law. This shows that the bottleneck is not in the architecture; rather it's the inductive bias. Additionally, the new symbolic regression results described above (which are in Appendix F) show that the state pretrained model correctly recovers the correct law over different datasets, while the next-token model --- which has the same architecture --- results in different laws across datasets.
> > > >
> > > > > _In a few places (mostly the introduction), the authors mention they study world-modelling capabilities along two distinct dimensions: “inductive bias towards state” and “recovering of state”. Of course, these terms refer to Definitions 2.1 and 2.2, but their intuitive meaning was not at all clear to me from reading just their short descriptions in the introduction._
> > > >
> > > > Great suggestion. We've updated the introduction to make these definitions clear, and our revisions to the Framework section based on your suggestions have also improved the clarity. We really appreciate the language you've suggested.
> > > >
> > > > > _“New functions” are simply “the learned model”, and should better be called that. It seems unnecessary to introduce the variable L, since m\hat ( · ; · ) can already be seen as the learning algorithm._
> > > >
> > > > Thank you for the suggestions. We agree that "learned model" is clearer than "new function" and we've updated the paper to incorporate this.
> > > >
> > > > As you suggest, given that we already have notation m\hat(.;D) it is unnecessary to introduce separate notation L for the learning algorithm. We have therefore dropped this extra notation in the revised draft.
> > > >
> > > >
> > > > > _The variable you introduce for the state space is /Phi, but you later use \mathcal{Q}.. and this should be made explicit in their names (for example, “inductive bias towards state /phi”). The quantitative metrics should be definitions, since they are more central to the paper (or non-trivial) than the binary definitions 2.1 and 2.2._
> > > >
> > > > Thank you for these writing suggestions relating to states. We've revised the framework section in light of these suggestions, and it's easier to read as a result.
> > > >
> > > > > _A large value of IB(Q), that is, a high “inductive bias towards state”, indicates that the learned model cannot be well-approximated as decomposing through state. This is intuitively backwards_
> > > >
> > > > We agree that the name for the inductive bias metric implies higher should be better, so we've followed your suggestion and redefined it to be the other way.
> > > >
> > > > > _I understand that the function /phi is supposed to be the “real” state function throughout, meaning the one actually used to generate the data. This is especially obvious in the experimental sections, but should somewhere be mentioned explicitly... In a couple places, you introduce the Orbital mechanics setting with the sentence “The true world model is the world”, meaning, of course, that the data is generated from Newton’s law... But I find this sentence doesn’t add anything_
> > > >
> > > > We appreciate your suggestion about being more clear about the state function, especially early on. We've updated the paper by:
> > > > - Stating that learners are applied to data from the real state function earlier in the paper.
> > > > - Replacing "the true world model is the world" with more information about the state function in Newtonian mechanics.
> > > >
> > > > > _Probably the footnote of Figure 1 should quickly mention how these laws are extracted from model behavior, which is, of course, through symbolic regression..._
> > > >
> > > > Thank you for the thorough writing suggestions. We've incorporated these into our updated revision and we are glad that it reads more smoothly as a result.

---

> > ### Comment · Reviewer_PEo1 · 2024-11-23
> > **Response**
> >
> > I'm thankful that the authors have reciprocated my in-depth review with an in-depth response, including not only stylistic revisions, but also some new experiments which certainly improve the reader's understanding of the paper's subject. Still, some reservations remain, so I include some detailed comments below.
> >
> > The new correlational metric is very pertinent and sensible. If anything, I am surprised that a metric of this flavor didn't play a more prominent role in the paper earlier on, which would have allowed for more iteration and experimentation. Especially so taking into account that this metric does importantly take advantage of the paper's focus on a learning algorithm (instead of a single model), unlike the previous ones, more on that below.
> >
> > It is good that the results for this new metric tend to point in the same direction as the previous ones, albeit not overwhelmingly for all architectures, if I'm reading these tables correctly.
> >
> > As a small and unimportant stylistic note, the rigid wording of lines 930-35 is a bit weird. You could simply say that you want this set to include datapoints from equal as well as differing states.
> >
> > I like the explicit explanation, provided in the authors' rebuttal, about what the main contributions of this paper are and aren't, the experiments' two key purposes. I would have preferred to see this explicitly present somewhere in the paper.
> > I thank the authors' added short remarks to Section 4.3. I would still have preferred the paper to give more centrality to the follow-up study of this phenomenon (which seems to me more vividly interesting than other parts which make up more of the length of the paper), although of course it is too late for this.
> >
> > I appreciate that the authors have implemented some major changes to their Framework section so closely in line with my previous comments. I will respectfully remark, though, that I am somewhat alarmed that such relatively obvious mathematical points had not been observed prior to submission, especially in a paper whose main contribution are the binary and non-binary metrics presented therein.
> >
> > In their rebuttal, and in response to my point that their metrics just seem to "average across individual models/datasets" (and thus, the learning algorithm doesn't add much), the authors remark that this is true of IB(), but not of SR(), which is computed across datasets. While it's true that SR() is not just, literally, averaging a value over datasets, I'd still say that what SR() does (fitting a state-function r() to all models instead of the ground truth, and checking its error relative to ground truth) is not too interesting or intuitive a way of taking advantage of the learning algorithm dimension. I think the correlational metrics do that more naturally, which unfortunately don't have a more central role in the paper.
> >
> > I feel somewhat conflicted as whether to increase my score, especially given how much the authors have added to the paper. But as I've mentioned in some of the above comments, many of these additions seem like substantive revisions to its content. Indeed, their earlier inclusion would have allowed the authors to construct a more robust paper around them. Because of this consideration and some remaining worries around the substantiveness of the contribution, and despite my many thanks to the authors for their engagement, I am maintaining my score.

---

> > > ### Author Response · Authors · 2024-11-24
> > >
> > > Thank you for the positive feedback about the new experiments, and we appreciate your continued thorough engagement with our work. We're grateful for how your feedback has helped improve the paper's clarity and technical foundations.

---

### Official Review · Reviewer_s8bn · 2024-11-02

**Soundness:** 3
**Presentation:** 3
**Contribution:** 3
**Rating:** 6
**Confidence:** 2

**Summary:**

This paper proposes a new framework to evaluate whether the learning algorithms utilizes world models to make predictions. Basically, it develops two metrics that captures the model's world modeling capabilities, one is inductive bias towards world state when it transfers into new tasks; the other is the degree to which a learner recovers state, such as whether the new prediction function utilizes information of all the state, or some aspects of them. Experiments are done under controlled setup, the results are kind of interesting, it shows that although most of the models perform well on next-token predictions, they have poor inductive bias towards world state, hence transfer properties, and the authors hypothesized that the models might rely on bundles of heuristics.

**Strengths:**

- The paper is well-written and easy to follow. The evaluation framework is described in a clear manner, from the definition of inductive bias and state recovery, towards on how to measure them via the lens of transfer learning, etc.
- This paper does show some valuable insights on whether or not the models are utilizing world models to do the prediction, I do find the empirical observations about the model's limited inductive bias and poor transfer properties interesting. For example, the case listed in Bundles of Heuristics about the model does not need to understand the board correctly in order to make the legal moves is kind of surprising.
- The controlled synthetic experiments is well-done and it shows very clear evaluation and understanding on the world-modeling capabilities of current models that good at next-token prediction.

**Weaknesses:**

- I’m curious whether the measurement of inductive bias is highly dependent on the distribution of the training data. For instance, if the data distribution is more uniform and not covers only a small subspace, would this significantly alter the conclusions? In the case of orbital mechanics, would a model trained on a more uniform distribution still capture universal laws?
- In the cases of Lattice and Othello, do the ground-truth states represent the minimal information needed for next-token prediction? I’m particularly considering the results shown in Figure 2.
- I appreciate the 'empirical' evaluation framework, but it seems to depend on various factors, like the IB loss function, reconstruction loss, transfer setup, data distribution, and so on. How sensitive are the conclusions in the paper to these parameters, and how well do they generalize?
- Could you comment more on how these world models connect with the memorization and reasoning concepts in the LLM community if possible?

(Note: I’m not an expert in world models; I’m reading this from a general perspective.)

**Questions:**

See weakness.

---

> ### Author Response · Authors · 2024-11-21
> **Author Rebuttal**
>
> Thank you for your thoughtful review of our paper and we’re glad you found our results interesting and surprising. We appreciate the positive feedback, and your suggestions have helped improve the paper.
>
> > _I’m curious whether the measurement of inductive bias is highly dependent on the distribution of the training data. For instance, if the data distribution is more uniform and not covers only a small subspace, would this significantly alter the conclusions? In the case of orbital mechanics, would a model trained on a more uniform distribution still capture universal laws?_
>
> This is a great question about the distribution of training data. We empirically study this in Othello and orbital mechanics:
> - Othello: We test models on two kinds of Othello datasets -- transcripts from real championship games (championship Othello), and those that come from random gameplay (synthetic Othello). These cover two kinds of training data distributions -- more uniform versus narrower. We don't see a significant difference in inductive bias but we find that state recovery is larger for championship Othello.
> - Orbital mechanics: We tried to get as much coverage in the training distribution as possible by, for instance, generating orbits with 10 distinct uniformly-spaced eccentricities and by randomly sampling the masses of the two objects and the initial distance between the two objects. It’s an interesting question whether the strength of inductive bias is dependent on the coverage of the data distribution, but our experiments show that despite the wide coverage, the trained model consistently shows weak inductive bias.
>
> > _In the cases of Lattice and Othello, do the ground-truth states represent the minimal information needed for next-token prediction? I’m particularly considering the results shown in Figure 2._
>
> This is a great question. The answer is yes: the state structures of both lattice and Othello obey a DFA, so the ground-truth state represents the minimal information needed for next-token prediction. [1] shows that there is an if-and-only-if relationship: perfect next-token prediction is necessary and sufficient for recovering ground truth states. But at the same time, near-perfect next-token prediction does not imply near-perfect state recovery, hence the importance of additional metrics.
>
> [1] Vafa, Keyon, et al. “Evaluating the World Model Implicit in a Generative Model.” NeurIPS. 2024.
>
> > _I appreciate the 'empirical' evaluation framework, but it seems to depend on various factors, like the IB loss function, reconstruction loss, transfer setup, data distribution, and so on. How sensitive are the conclusions in the paper to these parameters, and how well do they generalize?_
>
> This is a good question. We provide ablations across settings in Appendix D of the submission and find that the results are robust against these choices. Additionally, we've included new complementary evaluation metrics in Appendix G. These new evaluation metrics are based on correlation measures, so they don't require making choices about things like model architecture. Importantly, they come to similar conclusions as the original metrics in the main text.
>
> > _Could you comment more on how these world models connect with the memorization and reasoning concepts in the LLM community if possible?_
>
> This is an interesting question. Just as we found models can achieve good performance through "bundles of heuristics" rather than true world models, LLMs can generate valid outputs through memorization and pattern matching without deeper understanding. However, true reasoning capabilities likely require having a world model -- compressed, generalizable representations that can be applied to new situations. This is similar to how we find that models without strong inductive bias toward state struggle with transfer learning, similar to how LLMs that rely primarily on memorization often struggle with novel reasoning tasks. While memorization can enable good performance on tasks similar to training data, our metrics suggest that building AI systems capable of robust reasoning may require developing training approaches that encourage the formation of true world models rather than surface-level patterns.

---

### Official Review · Reviewer_qWH2 · 2024-11-03

**Soundness:** 3
**Presentation:** 2
**Contribution:** 3
**Rating:** 6
**Confidence:** 4

**Summary:**

The paper aims to test if a learner has a world model embodied in it.
To assess a learner’s world model, the paper measures its inductive bias when transferring to new tasks and proposes two metrics to measure the inductive bias: inductive bias (IB) and state recovery (SR).

Experiments on five pretrained models (RNN, LSTM, Transformer, Mamba, Mamba-2) in areas where the true world model is known (orbital mechanics, lattice problems and Othello games) show that these models learn bundles of heuristics that enable them to perform well on next-token prediction despite preserving little of state and having poor transfer properties for new problems.

**Strengths:**

1. The paper proposes a new procedure to test if a learner has a world model embodied in it by measuring its inductive bias when transferring to new tasks, instead of studying the behavior of fixed models.
This procedure provides a novel perspective to investigate whether pretrained models develop world
models.
2. The test models and scenarios in the experiments are representative, and results are inspiring.

**Weaknesses:**

1. Experiment results may not generalize to large pretrained models, such as LLMs, which are pretrained in a vast amount of data and have billions of parameters. I wonder if authors have plans to scale up the experiments to larger models, or if they could discuss potential limitations in generalizing their findings to LLMs.
2. The title for Table 1 and Table 2 should be more detailed and include result analysis, including key trends that are highlighted in the tables and why the trend exits.

**Questions:**

1. Please provide a more explicit explanation of the question "what does it mean to test if a learner has
a world model embodied in it?", and how experiment findings relate to this question. And please state the motivation behind this question.
2. What are the advantages of the proposed measuring procedure over other studies to assess whether pretrained models develop world models (e.g., studying the behavior of fixed models)?

---

> ### Author Response · Authors · 2024-11-21
> **Author Rebuttal**
>
> Thank you for your review. We’re glad that you found our methodology novel and our results "inspiring". We appreciate the positive feedback, and your suggestions have helped improve the paper.
>
> > _I wonder if authors have plans to scale up the experiments to larger models, or if they could discuss potential limitations in generalizing their findings to LLMs._
>
> This is a great question about LLMs. Our metrics are applicable to any learning algorithm where ground truth state is known to the analys. Much of the prior world model literature has focused on simpler synthetic domains like board games [1-3]. These testbed settings are important for developing general metrics and assessing model capabilities; e.g. what can be said about a transformer's world modeling capabilities if it can't extrapolate in even simple domains? Having developed and validated these metrics on the testbed settings, we're eager to apply them to LLMs in future work.
>
> [1] Toshniwal, Shubham, et al. “Chess as a Testbed for Language Model State Tracking.” AAAI. 2022.
> [2] Li, Kenneth, et al. “Emergent World Representations: Exploring a Sequence Model Trained on a Synthetic Task.” ICLR. 2023.
> [3] Vafa, Keyon, et al. Evaluating the World Model Implicit in a Generative Model. NeurIPS. 2024.
>
> > _Please provide a more explicit explanation of the question "what does it mean to test if a learner has a world model embodied in it?", and how experiment findings relate to this question. And please state the motivation behind this question._
>
> Thank you for this question. Below we provide more background and we've updated the paper to include this point more clearly.
>
> While existing approaches often study whether a fixed model's outputs are consistent with a world model (e.g., checking if language models make physically plausible predictions), this doesn't tell us whether the model actually learns and transfers knowledge using the correct world model. The motivation for this question is both theoretical and practical:
> - Theoretical: Having a world model means more than just making valid predictions - it implies having a compressed, generalizable representation of how a domain works that can be applied to new situations. A true world model should manifest in how a system learns and adapts, not just in its static outputs.
> - Practical: Many of the benefits of world models (like few-shot learning and transfer) specifically arise from how models learn new tasks. For example, if a language model truly has a physics world model, it should learn new physics-related tasks by using its underlying physics knowledge rather than learning each task from scratch.
>
> Our experimental findings directly address this question by:
> - Providing a formal framework that defines what it means for a learner (rather than just a fixed model) to use a world model, based on whether it shows inductive bias toward using state information when learning new tasks
> - Demonstrating that models which appear to have world models based on traditional evaluations (e.g., making valid next-token predictions) often fail to show the learning patterns we would expect from a true world model
> - Revealing that many models rely on "bundles of heuristics" rather than coherent world models -- they can make valid predictions but don't learn new tasks by leveraging an underlying model of the domain
>
> The orbital mechanics example particularly illustrates this -- while the model can make accurate predictions about planetary motion, our metrics reveal it doesn't actually learn using Newtonian physics as its inductive bias. Instead, it learns different heuristics for different situations, showing it lacks a true physics world model despite superficially valid outputs.
>
> > _What are the advantages of the proposed measuring procedure over other studies to assess whether pretrained models develop world models (e.g., studying the behavior of fixed models)?_
>
> Both fixed models and learners are important to study. In addition to the above, another reason we focus on learners is because one of the promises of foundation models having world models is that they can be adapted to new tasks by using world models. In other words, having reliable world models means algorithms can build on top of each other. The physics example illustrates why failures of world models in learners can be damaging; because the learner doesn't use Newtonian physics, it will be brittle for adapting to other tasks that rely on it.
>
> > _The title for Table 1 and Table 2 should be more detailed and include result analysis, including key trends that are highlighted in the tables and why the trend exists._
>
> Thank you for your suggestion for the captions for Tables 1 and 2. We've updated them to include more analysis.

---

> ### Comment · Reviewer_qWH2 · 2024-11-25
>
> Thanks for the authors' response. My concerns have been addressed. And the explanations above are expected to be included in the final version of the paper. One more suggestion is to improve the readability of current version, especially for those who don't have relevant background. I've increased my confidence score.

---

> > ### Author Response · Authors · 2024-11-25
> >
> > We're glad your concerns have been addressed. We've updated the paper to include the explanations above and we'll continue to improve the readability for the next revision. Thank you for the positive feedback and for engaging with our work.

---

### Official Review · Reviewer_ZvCa · 2024-11-03

**Soundness:** 1
**Presentation:** 1
**Contribution:** 1
**Rating:** 3
**Confidence:** 3

**Summary:**

This paper introduces a framework for evaluating whether learning algorithms develop world models using two measurements: i) inductive bias toward “state” and ii) state recovery. These metrics are used across various model architectures and a handful of domains to highlight that high performance on next-token prediction does not imply inductive bias toward state.

**Strengths:**

1. **Originality**. I think the overall question is interesting.

2. **Quality**. I like that the authors choose simple testbeds to investigate potentially interesting phenomenon.

3. **Clarity**. I think the structural choice of first introducing the theoretical framework then moving to the exact implementation makes sense.

4. **Significance**. The paper provides framework for understanding why models might perform well on pretraining tasks without developing true world modelsI. I like that the paper includes evaluations of multiple modern architectures, and this investigation appears to reveal interesting patterns. For example, Mamba models perform better with state supervision but worse with next-token pretraining vs. transformers (Table 1).

**Weaknesses:**

1. **Lack of clear, grounded definitions.** The paper defines a world model as a mapping from inputs to “state” (page 1); however, this mapping is clearly insufficient. Without a precise, formal definition of what a world model is and ought to achieve, the two metrics for measuring world model quality are not well justified. For example, a lookup table mapping inputs to states would satisfy this definition but isn't what we typically mean by a "world model". As another example, a useful world model might abstract away irrelevant state details while capturing key dynamics. I suggest i) a more precise definition of a good world model and its properties and ii) a discussion of how these properties relate to the extensive world model literature. This lack of definition is confusing given existing definitions in the literature [1-3].

2. **Confusing writing.** Continuing with W1, the paper uses terms like "state" without grounding in established frameworks (e.g., MDPs). This lack of clarity made the paper very hard to follow. I also found it difficult to understand the motivation and premise behind the work.


3. **Lack of engagement with relevant literature.** Given the usage of reinforcement learning terminology, I am surprised by the lack of engagement with the relevant literature. For example, this work does not engage with the extensive literature on state abstractions, which seeks to address similar research questions. The state abstraction literature in reinforcement learning [4-7] has extensively studied questions like: i) What makes a good state representation?, ii) When can we compress states while preserving important properties?, and iii) How do different types of state abstractions affect learning and transfer? The "partial reconstruction of state" metric, defined in Definition 2.2, seems related to concepts like bisimulation metrics (how can we cluster states that behave similarly?) and MDP homomorphisms (what state mappings preserve the essential dynamics?).

4. **Unnecessary “mathiness”**. Proposition 2.3 is tautological.

5. **Unclear methodological decisions**. I don’t understand the synthetic dataset construction part in Section 2.2. If one constructs a dataset that is generated from some behavior policy, there is no reason that assigning random outputs would map to any meaningful task.


6. **Results presentation**. Results lack significance testing or even any error bars.

7. **Wrong format**. The paper is missing the line numbers that are standard in the ICLR template.

[1] Ha, David, and Jürgen Schmidhuber. "World models." arXiv preprint arXiv:1803.10122 (2018).

[2] Chen, Chang, et al. "Transdreamer: Reinforcement learning with transformer world models." arXiv preprint arXiv:2202.09481 (2022).

[3] LeCun, Yann. "A path towards autonomous machine intelligence version 0.9. 2, 2022-06-27." Open Review 62.1 (2022): 1-62.

[4] Li, Lihong, Thomas J. Walsh, and Michael L. Littman. "Towards a unified theory of state abstraction for MDPs." AI&M1.2 (2006): 3.

[5] Ravindran, Balaraman, and Andrew G. Barto. "Approximate homomorphisms: A framework for non-exact minimization in Markov decision processes." (2004).

[6] Van der Pol, Elise, et al. "Mdp homomorphic networks: Group symmetries in reinforcement learning." Advances in Neural Information Processing Systems 33 (2020): 4199-4210.

[7] Abel, David, David Hershkowitz, and Michael Littman. "Near optimal behavior via approximate state abstraction." International Conference on Machine Learning. PMLR, 2016.

**Questions:**

1. Could the authors please clarify and motivate this instantiation of a world model?

2. Could the authors please justify why these choices of metrics are connected to desirable properties of a world model?

3. I don’t think the finding that incorrect board position still yields correct predictions of legal moves is that surprising from the state abstraction perspective. What seems to be happening is that the internal learned representation is lossy; model-based RL typically works with imperfect models that learn sufficient dynamics for good policy learning. In essence, one doesn't need a perfect world model for decision making. Could the authors clarify why this is undesirable?

---

> ### Author Response · Authors · 2024-11-21
> **Author Rebuttal (1/2)**
>
> Thank you for your review. Your review makes several helpful points that will improve our paper.  However, our rebuttal clarifies a potential confusion about differences between the study of world models in the RL and LLM literatures. We've made this more clear in our revision.
>
> > _The paper defines a world model as a mapping from inputs to “state” (page 1); however, this mapping is clearly insufficient... Given the usage of reinforcement learning terminology, I am surprised by the lack of engagement with the relevant literature... The "partial reconstruction of state" metric, defined in Definition 2.2, seems related to concepts like bisimulation metrics (how can we cluster states that behave similarly?) and MDP homomorphisms (what state mappings preserve the essential dynamics?)... Could the authors please clarify and motivate this instantiation of a world model?_
>
> We think there's a potential confusion: while the term "world models" is used in both the RL and LLM literatures, it means different things in each literature and the goals are quite different. While your comments refer to the concept of world models in RL, our paper is contributing to the LLM literature that evaluates whether LLMs build world models that accord with the real world [1, 2, 3, 4]. In RL, world models refer to representations (or even the specific neural network) learned by an agent, and their quality is typically measured by how well they perform policy optimization [5, 6]. In contrast, the LLM literature focuses on _evaluation_: a learning algorithm is evaluated by how well it can recover an externally defined mapping from a true world model. In other words, it seeks to answer the question, “does a model's behavior reflect understanding of the true world state?” Our work follows this evaluation perspective in the LLM literature, and we contribute to this literature by studying learning algorithms rather than fixed models.  Similarly, despite using similar terminology, the state abstraction literature in RL (e.g., bisimulation, MDP homomorphism), which is primarily focused on aggregating and simplifying the state-space of an MDP [7, 8], has a different goal from that of world model evaluation in LLMs. Again, by contrast, the LLM literature focuses on assessing whether generative models recover underlying world models with high representational fidelity.
>
> We agree that an arbitrary lookup table recovered by a model is not an interesting world model. This is why the LLM literature studies whether learning algorithms can recover real-world mappings. What makes mappings from inputs to states interesting in the real world is that we can build on top of them; for example, in the physics example, the mapping from a sequence of planetary movement to a state vector is scientifically useful because those states (e.g. relative velocity and masses) are important concepts for other physical tasks.
>
> We've updated the related work to include a discussion about the differences in world model recovery in LLMs and RL. Please let us know if you have any more questions or if anything is unclear about the LLM world model literature.
>
> [1] Toshniwal, Shubham, et al. “Chess as a Testbed for Language Model State Tracking.” AAAI. 2022.
> [2] Li, Kenneth, et al. “Emergent World Representations: Exploring a Sequence Model Trained on a Synthetic Task.” ICLR. 2023.
> [3] Li, Belinda Z., et al. “Implicit Representations of Meaning in Neural Language Models.” ACL. 2021.
> [4] Vafa, Keyon, et al. “Evaluating the World Model Implicit in a Generative Model.” NeurIPS. 2024.
> [5] Ha, David, and Jürgen Schmidhuber. “World Models.” 2018.
> [6] Chen, Chang, et al. “TransDreamer: Reinforcement Learning with Transformer World Models.” arXiv:2202.09481. 2024.
> [7] Abel, David, et al. Near Optimal Behavior via Approximate State Abstraction. arXiv:1701.04113. 2017.
> [8] Li, Lihong, Thomas J. Walsh, and Michael L. Littman. "Towards a unified theory of state abstraction for MDPs." AI&M1.2 (2006): 3.
>
>
> > _I don’t understand the synthetic dataset construction part in Section 2.2. If one constructs a dataset that is generated from some behavior policy, there is no reason that assigning random outputs would map to any meaningful task._
>
> The synthetic datasets are constructed to study the extrapolation behavior of learners. Importantly, they're not completely random outputs. They're constructed to enforce that _within a dataset, two points in the same state have the same mapping_. It's just the assignments of points that's random. It's exactly because there's no structure beside state that enables us to test whether learners extrapolate based on state. By repeatedly applying a learner to different datasets, we can test whether its extrapolations can be predicted based on extrapolations of other points with the same state. We've made this more clear in the revision.

---

> > ### Author Response · Authors · 2024-11-21
> > **Author Rebuttal (2/2)**
> >
> > > _I don’t think the finding that incorrect board position still yields correct predictions of legal moves is that surprising from the state abstraction perspective. What seems to be happening is that the internal learned representation is lossy; model-based RL typically works with imperfect models that learn sufficient dynamics for good policy learning. In essence, one doesn't need a perfect world model for decision making. Could the authors clarify why this is undesirable?_
> >
> > As discussed above, we're studying this question not from the RL perspective but from the LLM world model literature perspective. There are a few reasons why this is undesirable for LLMs:
> > An LLM's world model could potentially tell us something new about the world. This is one reason LLMs are being trained in domains like genetics, chemistry, and protein folding [9, 10, 11, 12, 13]. If LLMs aren't actually building complete world models, but rather only building enough to satisfy next-token prediction, they will not be useful in these scientific settings that aim to uncover larger theories.
> > Another promise of LLMs building world models is that we can build on top of them. E.g. if an LLM builds a world model of the map of Manhattan, we could compose it with search algorithms like breadth-first search [4]. If models are not building correct and compact world models, but only enough to satisfy next-token prediction, it leaves them brittle for building up.
> >
> >
> > [9] Jablonka, K.M., Schwaller, P., Ortega-Guerrero, A. et al. Leveraging large language models for predictive chemistry. Nat Mach Intel.
> > [10] Boiko, D.A., MacKnight, R., Kline, B. et al. Autonomous chemical research with large language models. Nature.
> > [11] Chowdhury, R., Bouatta, N., Biswas, S. et al. Single-sequence protein structure prediction using a language model and deep learning. Nat Biotechnol.
> > [12] Zeming Lin et al., Evolutionary-scale prediction of atomic-level protein structure with a language model. Science.
> > [13] G. Benegas, S.S. Batra, Y.S. Song, DNA language models are powerful predictors of genome-wide variant effects, Proc. Natl. Acad. Sci. U.S.A.
> >
> > > _Proposition 2.3 is tautological._
> >
> > Proposition 2.3 is intended to be a summary of the previous section, and so we agree that it doesn't provide new information. We've updated the paper by removing "proposition" from the heading and fleshing out the summary.
> >
> > > _Results lack significance testing or even any error bars._
> >
> > We've added error bars to the new experiments we perform in Appendix G (Tables 7-10). While we didn't have time during the rebuttal period to re-run all experiments for error bar generation (our initial submission involved training more than 1000 models), our results for the new metrics point to the statistical significance of our results and we will include error bars for all experiments in the final revision.
> >
> > > _The paper is missing the line numbers that are standard in the ICLR template._
> >
> > Thank you for catching this. We've included the line numbers in the updated version.

---

### Author Response · Authors · 2024-11-21

We thank the reviewers for their careful evaluation and feedback. We're glad that you found our paper "inspiring" (qWH2) and offering "valuable insights" (s8bn). We're also pleased that you found our work to be "pertinent and of broad scientific interest" (PEo1). Your constructive feedback has been invaluable for improving the clarity and sharpening the main conceptual points of our work.


In response to reviewer's suggestions, we've updated the PDF with the following:
- **Additional metrics:** We've added new measures in Appendix G that are based on correlation and therefore provide more intuitive scaling. These measures track whether a learner's extrapolations are more correlated for data points in the same state versus different states. These results point to the same conclusions as the original metrics, showing the robustness of each result. See Tables 7-10.
- **New empirical results:** We've expanded our physics experiments with new symbolic regression analyses in Appendix F. This new section provides stronger evidence for the "piecemeal heuristics" hypothesis -- showing that models construct different laws for different sequences rather than learning a single incorrect law.
- **Revised framework section:** Our new section includes clearer mathematical definitions, simplified notation, and more intuitive terminology (e.g. renaming "inductive bias towards state" to "respects state" and clarifying state recovery concepts). We've also restructured the section to introduce quantitative metrics after establishing the core definitions.

These changes have made the paper stronger, and we thank you for suggesting them. We’ve also included more details about these results in our individual responses to each reviewer.

---

### Meta-Review · Area_Chair_HRMP · 2024-12-23

**Metareview:**

This paper proposes two quantities to characterize whether learners represent a model of the task (world, or environment). These two quantities are (i) inductive bias towards representing the true state of problem, and (ii) ability to recover the true state from the observations. One can think of these quantities as the rate and distortion respectively in information bottleneck. The paper develops a procedure to compute these quantities and argues that in many examples, pre-training methods do not learn such world-model relevant representations, they instead build upon “bundles of heuristics”.

There was substantial discussion during the review period. I would like the thank both the authors and the reviewers for engaging in the process and improving the original manuscript. While the motivations of the authors are clear, there are major questions around whether the two quantities proposed in the paper can completely characterize the research question.

One more point (which was not pointed out in the review process) is that a pre-trained loss leads to a representation that is sufficient to predict the future state. Such a representation need not be sufficient to reconstruct the full state (e.g., if some dynamical models of a system are not sufficiently excited), nor be sufficient for a downstream task (e.g., a policy that gets a high reward). In other words, pre-trained learners will learn “world models” only if all modes of a system are sufficiently excited in the training data (assuming that the representational capacity of the learner is not limiting). It is difficult to understand what one is to take away from the present manuscript, because the quantities explored in the paper and the findings cannot be general. For these reasons, I do not recommend this paper to be accepted.

**Additional Comments On Reviewer Discussion:**

Reviewer ZvCa had concerns about rigorousness of the definitions proposed in this paper, writing and ambiguity of the numerical experiments. I share all these concerns with the Reviewer. The authors have responded to these concerns in their rebuttal but it is not clear how their rebuttal helps the manuscript be more precise and rigorous.

Reviewer qWH2 commented on whether these findings will hold for very large language models. They also had a few clarifying questions. The authors have addressed these questions satisfactorily.

Reviewer s8bn had comments on how it is surprising that the model can predict legal Othello moves correctly without predicting the entire state correctly. They were concerned about how general these findings are.

Reviewer PEo1 had an elaborate review which revolved around the same points raised above. In particular, what do the metrics mean and why are they reasonable quantities to study. The reviewer also pointed out a number of technical considerations, e.g., how the definition of inductive bias towards the state is related to the notion of a sufficient statistic, and how the ability to recover the state from the observations is related to observability. There was an elaborate discourse following the original review, during which the authors substantially modified the paper.

Altogether, the comments of Reviewer ZvCa and PEo1, the author rebuttal, and my independent reading of the paper were responsible for my decision.

---

### Decision · Program_Chairs · 2025-01-22

Reject